# AI-Driven Detection of Obstructive Sleep Apnea Using Dual-Branch CNN and Machine Learning Models

**DOI:** 10.3390/biomedicines13051090

**Published:** 2025-04-30

**Authors:** Manjur Kolhar, Manahil Muhammad Alfridan, Rayan A. Siraj

**Affiliations:** 1Department of Health Information Management and Technology, College of Applied Medical Sciences, King Faisal University, Al-Ahsa 36362, Saudi Arabia; mmalfridan@kfu.edu.sa; 2Department of Respiratory Therapy, College of Applied Medical Sciences, King Faisal University, Al-Ahsa 36362, Saudi Arabia

**Keywords:** obstructive sleep apnea (OSA), dual-branch CNN, ECG, machine learning, deep learning

## Abstract

**Background/Objectives:** The purpose of this research is to compare and contrast the application of machine learning and deep learning methodologies such as a dual-branch convolutional neural network (CNN) model for detecting obstructive sleep apnea (OSA) from electrocardiogram (ECG) data. **Methods:** This approach solves the limitations of conventional polysomnography (PSG) and presents a non-invasive method for detecting OSA in its early stages with the help of AI. **Results:** The research shows that both CNN and dual-branch CNN models can identify OSA from ECG signals. The CNN model achieves validation and test accuracy of about 93% and 94%, respectively, whereas the dual-branch CNN model achieves 93% validation and 94% test accuracy. Furthermore, the dual-branch CNN obtains a ROC AUC score of 0.99, meaning that it is better at distinguishing between apnea and non-apnea cases. **Conclusions:** The results show that CNN models, especially the dual-branch CNN, are effective in apnea classification and better than traditional methods. In addition, our proposed model has the potential to be used as a reliable, non-invasive method for accurate OSA detection that is even better than the current state-of-the-art advanced methods.

## 1. Introduction

Obstructive sleep apnea (OSA) is an important sleep problem marked by repeated instances of full blockage of the upper airway while sleeping [1]. These occurrences disrupt the flow of oxygen in the individual, leading to heart-related issues such as blood pressure, irregular heartbeats, and a higher likelihood of heart failure and stroke [2]. Obesity is identified as a risk factor for OSA, underscoring the importance of a diagnosis in effectively addressing and treating OSA [3]. Traditionally, the diagnosis of OSA relies on polysomnography (PSG), an extensive and comprehensive sleep study that monitors multiple physiological parameters overnight [4]. While PSG is considered the gold standard for diagnosing OSA, it is resource intensive, time consuming, and often uncomfortable for patients [5]. With advancements in technology and machine learning, the electrocardiogram (ECG)-based detection of OSA offers several significant benefits over traditional methods: it is non-invasive, cost-effective, and scalable [6]. Unlike PSG, ECG-based detection is less intrusive and can be performed with minimal discomfort to the patient [7]. ECG devices are simple to use and do not require the extensive setup involved in PSG, making the diagnostic process more comfortable and patient friendly [8]. ECG devices are more affordable and accessible compared with the sophisticated equipment required for PSG [9]. This cost-effectiveness makes ECG-based detection a viable option for widespread screening and diagnosis, particularly in resource-limited settings [10]. ECG monitoring can easily be implemented on a large scale, making it suitable for mass screening initiatives [11]. This scalability ensures that a larger population can be screened for OSA, leading to early detection and intervention.

Advanced algorithmic detection methods, particularly those based on machine learning and deep learning, have shown great promise in identifying OSA from ECG signals [12]. These methods leverage specific ECG features associated with apneic events, such as heart rate variability, R–R intervals, P-wave and T-wave morphology, and QT interval prolongation [13]. Convolutional neural networks (CNNs) are highly effective in extracting features from raw ECG signals and identifying complex patterns [14]. They can automatically detect the relevant features that are indicative of OSA without the need for manual feature engineering. Recurrent neural networks (RNNs), particularly long short-term memory (LSTM) networks, are well-suited to handling sequential data such as ECG time series [15]. They can capture temporal dependencies and variations in the ECG signal associated with apneic events. Combining CNNs and RNNs in a hybrid architecture leverages the strengths of both models [16]. CNNs can extract spatial features [17], while RNNs can handle temporal dependencies, resulting in the more accurate detection of OSA. By utilizing these advanced algorithmic techniques, ECG-based detection systems can achieve high accuracy in diagnosing OSA. These systems can be integrated into wearable devices or home monitoring solutions, providing continuous and real-time assessments of sleep quality and the early detection of OSA. In summary, the ECG-based detection of OSA, enhanced by sophisticated algorithmic methods, offers a non-invasive, cost-effective, and scalable alternative to traditional PSG. This approach has the potential to transform the diagnosis and management of OSA, improving patient outcomes and reducing the burden on healthcare systems.

In this study, we developed a novel annotation method to enhance the accuracy of ECG-based OSA detection. We introduced a robust dual-branch CNN model specifically designed for this task, achieving superior performance compared to existing advanced methods. Our approach provides a reliable, accurate, and non-invasive diagnostic tool, showing strong potential for practical use in clinical settings. We then applied various machine learning algorithms, such as decision trees, random forests, and CNNs, to recognizing patterns in the ECG data that indicate apnea episodes. To further enhance the detection precision, we created a dual-branch CNN model that leverages layers to capture local and global features of the ECG signals. This dual-branch model underwent validation using a dataset generated using the synthetic minority over-sampling technique (SMOTE) to tackle the inherent class imbalance in the dataset. Our model exhibited excellent performance levels with accuracy rates and strong area under the receiver operating characteristic curve (ROC AUC) scores on both the validation and test datasets. We created representations including confusion matrices, ROC curves, and classification reports to evaluate the model’s effectiveness and elucidate the classification results and provide unambiguous visual and quantitative insights into the model’s performance across individual classes, as well as the mean and standard deviation obtained across the five-fold cross-validation procedure.

## 2. Literature Review

The authors of [18] developed a new annotation method aimed at improving the accuracy of the ECG-based detection of OSA. They introduced a robust dual-branch CNN model tailored for this purpose, which performed better than existing advanced approaches. Their technique provides a reliable, precise, non-invasive diagnostic solution with considerable potential for practical applications in clinical environments [18]. AIOSA is a deep learning model that detects OSA from heart and breathing signals. It achieves high accuracy and is compatible with data collected from hospitals and standard benchmarks. However, it is computationally expensive and sensitive to signal noise, and its functioning is not easily explainable, which poses a challenge to its use in clinical practice to some extent [19]. Other researchers attempted to determine whether KL-6 levels in the blood can reveal ‘occult’ lung injury in people with OSA by studying 197 patients; they observed that higher KL-6 levels are associated with more severe OSA [20]. The authors of [21] aimed to develop a user screening model for OSA using biochemistry markers and demographic information. Data from 2020 to 2023 were examined using four machine learning techniques, with logistic regression (LR) showing the performance. The LR model, including factors such as BMI (AUC = 0.699) and the glucose (TyG) index (AUC = 0.656), demonstrated results in the training (AUC = 0.794; F1 score = 0.841), validation (AUC = 0.777; F1 score = 0.827), and testing (AUC = 0.732; F1 score = 0.788) groups. This model serves as a tool for healthcare professionals to assess the need for PSG in suspected OSA patients. The model’s performance is quite good; however, it has not been validated in any patient group other than the one used in the original study, meaning that confidence in its usefulness in other populations or healthcare contexts is limited.

The authors of [22] created an automated deep learning system to identify OSA events by simultaneously analyzing ECG and thoracic movement signals. The model was built using the ResNeSt34 algorithm and data from 420 PSG cases. The combined signals approach outperformed the ECG method, achieving an accuracy of 89.0%, precision of 88.8%, recall of 89.0%, an F1 score of 88.2%, and an AUC of 92.9%. In comparison, the ECG model had an accuracy of 84.1%, precision of 83.1%, recall of 84.1%, an F1 score of 83.3%, and an AUC of 82.8%. These findings highlight the effectiveness of detecting OSA events when thoracic movement signals are incorporated. The authors of [23] evaluated the Obstructive Sleep Apnea Smart System (OSASS), which uses a novel multi-resolution residual network (Mr-ResNet) to diagnose and classify OSA based on single-channel nasal pressure airflow signals. Data from sleep centers were used to develop and test the model. In the primary test set, Mr-ResNet achieved sensitivity, specificity, accuracy, and an F1-score of 90.8%, 90.5%, 91.2%, and 90.5%, respectively. OSASS showed strong agreement with registered polysomnographic technologists’ (RPSGTs’) scores, with Spearman correlations of 0.94 and 0.96 and Cohen’s Kappa scores of 0.81 and 0.84. The findings suggest that OSASS could be a valuable tool for clinical OSA diagnosis and classification. The authors of [24] explored the use of 3D maxillofacial shape analysis combined with machine learning (ML) to predict OSA in 280 Caucasian men. Compared with traditional questionnaires (BERLIN and NoSAS), the ML analysis of 3D craniofacial shapes demonstrated higher specificity (56%) and good sensitivity (80%) for detecting OSA (AHI ≥ 15 events/h). The performance further improved (ROC = 0.75) when combined with patient anthropometrics. This approach is proposed as a rapid, efficient, and cost-effective OSA screening tool, offering a promising alternative to traditional PSG methods. The authors investigated the automatic detection of OSA events using a deep learning approach: specifically, CNN. The model analyzes blood oxygen saturation, oronasal airflow, and ribcage and abdomen movements to classify sleep apnea events with 1 s annotations. Using the PhysioNet Sleep Database, the model achieved an average accuracy of 79.61% across normal, hypopnea, and apnea classes, demonstrating the potential of CNNs in accurately detecting OSA events in real time [25]. The authors of [26] analyzed the severity of OSA using resting-state EEG data without relying on PSG signals. The study examined EEG recordings from 25 OSA patients, focusing on features derived from delta, theta, and alpha sub-bands. Using ML techniques, the study identified combinations of features using the relief method and identified 15 features. The K Nearest Neighbors classifier (with K = 5) demonstrated excellent performance metrics, achieving an accuracy of 93.33%, sensitivity of 92.30%, specificity of 94.14%, and an AUC value of 0.98. These results indicate that using resting-state EEG may be an efficient method for evaluating OSA severity. The authors of [27] aimed to create a computer program using artificial intelligence to recognize patterns of sleep apnea in ECG signals from a lead. The model EfficientNet was used to detect apnea by analyzing data from 1656 patients, and the best outcomes (AUC = 0.917; accuracy = 0.855) were achieved by adjusting the slicing and sample weight settings. When paired with XGBoost to screen patients with AHI  >  30, the system reached an AUC of 0.975 and an accuracy of 0.928. Experiments conducted on PhysioNet data validated the model’s effectiveness in identifying levels of sleep apnea. AHI regression showed agreement (intraclass correlation coefficient > 0.9) with OSA severity classifications, achieving accuracies of 74.51% and 62.31% for the four class categories, along with Cohen’s Kappa values of 0.6231 and 0.4495 in the CHAT and private datasets. The diagnostic accuracies for AHI cutoffs (1, 5, and 10 events/h) were consistently above 84%. Analysis using Grad CAM heatmaps indicated that the model focuses on AF cessations and SpO2 drops to identify apneas and hypopneas with desaturations while often overlooking patterns of hypopneas associated with arousals. Therefore, employing a CNN + RNN model to assess AF and SpO2 could serve as a diagnostic tool for symptomatic children at risk of OSA [28].

Many recent studies employ small manual or semi-automated annotations and use standard signal preprocessing and augmentation techniques to complement the data. For instance, good accuracy has been achieved using standard CNN models; however, they have shown restricted generalizability to unannotated regions with dataset-specific annotations and poor signal quality. Furthermore, previous research on machine-generated QRS annotations often involved inaccuracies that required manual correction, which may have compromised the reliability of the models. Our study makes key contributions, improving apnea detection from ECG signals. We use dual-branch CNN architecture to train both broad and fine-grained ECG signal features simultaneously, making this model more robust than CNN models. Furthermore, our rigorous preprocessing pipeline entails sophisticated bandpass filtering, stringent segment normalization, and outlier detection to clean and stabilize the input data. Furthermore, we propose a new algorithm for the annotation of ECGs by simply counting the number of QRS complexes in an ECG segment, which increases the annotation accuracy and decreases dependence on inaccurate machine-generated labels. Our method is unique in that it clearly separates the training, validation, and testing processes to produce clinically meaningful and accurate results. We hereby propose collective enhancements as the novelty of our study, which distinguishes it from other studies (refer to Table 1).

Our approach addresses critical gaps identified in previous studies on ECG-based apnea detection. Our proposed dual-branch CNN architecture addresses the generalizability issues of conventional CNN models by extracting both general and specific ECG features which enhance model performance across different signal quality levels. Our novel QRS-count-based annotation algorithm improves label accuracy and decreases dependence on flawed machine-generated data. Our approach differs from previous work because we use a strict preprocessing strategy that divides the data into training, validation and test sets and only applies augmentation to the training data. This ensures unbiased evaluation and clinically meaningful results. Our preprocessing pipeline incorporates advanced filtering, normalization, and outlier detection to further stabilize the model input. Collectively, these enhancements contribute to a more scalable, interpretable, and reliable framework for detecting OSA from ECG signals, setting our method apart from traditional deep learning approaches in terms of both accuracy and clinical relevance.

## 3. Methods and Material

The study approach is outlined in Figure 1. It involves selecting the apnea ECG dataset, preparing the ECG signals, identifying features, and categorizing the data using ML and DL models.

### 3.1. Dataset

The dataset consists of 70 ECG recordings split into a training set of 35 records and a test set of 35 records. Each recording spans from 7 to 10 h in duration. It contains ECG signals, apnea annotations (only for the training set), and QRS annotations. In addition, eight records include signals (chest and abdominal effort oronasal airflow) and oxygen saturation (SpO2) data. This extensive dataset is designed for the development and testing of methods for apnea detection, with files provided to aid in the examination of respiratory and ECG signals. It is worth noting that certain QRS files may contain errors that need correction prior to analysis. Human-generated labels are created by experts to show whether apnea is present on a minute-by-minute basis (only accessible for the training dataset) [29]. The dataset offers two types of annotations for the computers in the Cardiology Challenge 2000, which are distinguished by the file extensions. apn. and .qrs. The .apn files include apnea annotations for the training set comprising 35 records established through expert assessments of respiration and oxygen saturation signals. Each annotation specifies whether apnea was ongoing at the beginning of each minute. The .qrs files, generated using the sqrs125 tool, are machine-generated annotations for QRS complexes. The rdann tool can also be employed to convert these annotation files into text for further examination.

### 3.2. Preprocessing

The initial steps outlined in this study are carefully crafted to prepare, standardize, and harmonize the ECG data for training an AI model. The process starts with bandpass filtering, which effectively eliminates any disturbances from the ECG signals (refer to Figure 2). Next, the ECG signal is divided into segments to facilitate processing and analysis by the model. Normalization is then implemented to ensure the uniformity of data across all segments, a factor for model performance (Figure 3). Points in the ECG signal that stand out are considered outliers, showing deviations from the amplitude. These outliers could suggest noise interference or unusual cardiac activities. It is essential to recognize and address these anomalies to guarantee ECG examination and interpretation.

### 3.3. Data Augmentation

To make the CNN model more robust at predicting apnea events from ECG signals, data augmentation methods were employed. These techniques involved exposing the model to a range of signal patterns, such as simulating heart rates through time stretching and mimicking sensor sensitivities by adjusting the amplitude scaling. Sensor noise was replicated using Gaussian noise, and variability in recording lengths was introduced through cropping. By incorporating these strategies, the training data became more varied, resulting in improved performance when processing ECG signals in real-world scenarios.

Our approach was intended to prevent information leakage; therefore, we divided the dataset into training, testing, and validation sets prior to preprocessing and augmentation. Data augmentation was applied to the training dataset only to address class imbalance, while the test and validation sets were not augmented. These settings enabled us to calculate the classification threshold using predictions from the training set and then apply them to the test and validation sets. Therefore, the model guarantees unbiased and reliable performance metrics by keeping the datasets strictly separate and using them only for their intended purposes to prevent any leakage of information that could artificially inflate results.

### 3.4. Annotation

Annotations played a role in training and validating models that identified sleep apnea from ECG signals accurately. The dataset comprised 70 records split into a training set and a test set, each containing digitized ECG signals with annotations. Human experts created apnea annotations, with respiration and related signals serving as the reference for detecting apnea events. These annotations were essential for ensuring the models’ precise detection of sleep apnea, thus enhancing their accuracy and clinical significance. While the machine-generated QRS annotations had errors, the apnea annotations helped rectify these errors, ensuring the model’s reliability and robustness. Leveraging these annotations enabled the researchers to train and evaluate their algorithms effectively, resulting in clinically relevant methods for detecting sleep apnea. The authors developed an algorithm for annotation; the .apn files consist of annotation data in format with annotations for each minute of every recording to show the presence or absence of apnea during a specific period. These files are accessible for the 35 recordings in the learning set. The .qrs files are annotation files generated by a machine using sqrs125. They are included for the convenience of individuals who prefer not to utilize their QRS detectors [29].

Algorithm 1 analyzes the QRS annotations from ECG signals at time intervals to identify instances of sleep apnea. It tallies the QRS complexes in each interval and checks them against a threshold. If the count is lower than the threshold, the interval is marked as apnea (‘A’); if not, it is labeled as no apnea (‘N’).
**Algorithm 1.** Annotating sleep apnea events based on QRS complex counts in ECG signalsInputqrs_annotations (sample indices)sampling_rate (samples per second)time_window (duration of each window in seconds)QRS_threshold (minimum QRSD count threshold to avoid labels)Output:apnea_annotations (list of labels: Apnea and normal lables)Step:1. window_size = time_window * sampling_rate2. Determine num_windows = len(qrs_annotations)//window_size  3. Initialize an empty labels list called apnea_annotations = []4. For each window i from 0 to num_window −1, do as follows:   4.1 Set time start_sample = i * window_size   4.2 Set end_sample = start_sample + window_size    4.3 Count QRS in the current time window within start_sample, end_sample   4.4 if QRS_count < QRS_threshold , then    4.4.1 append apnea_annotations.append((‘A’, start_sample))   Else    4.4.2 apnea_annotations.append((‘N’, start_sample))

### 3.5. Proposed Solutions

#### 3.5.1. CNN Model

Figure 4 provides an overview of the layers in the CNN model, including their output shapes, parameter counts, and connections.

The CNN model illustrates the dimensions of the output number of parameters and the connections between them. It includes layers, max pooling layers, batch normalization, a flattening layer, connected layers, and a final dropout layer that results in a singular output for binary classification. Our model, used in this study, has an organized design that combines layers and pooling layers, resulting in fully connected layers. Notably, the pooling layers are strategically placed after every layer. The network consists of layers followed by max pooling layers to reduce data dimensions. The output from the pooling layer is transformed into a vector for input into fully connected layers for classification purposes. To address overfitting concerns, batch normalization layers were incorporated along with dropout layers following max pooling and connected layers at a dropout rate of 0.5. The ReLU activation function was consistently used throughout the model, except in the output layer, which employed a sigmoid activation function for classification tasks. For optimization strategies, we opted for the Adam optimizer with a learning rate that was fine-tuned during training sessions. The learning rate was established at 0.001 based on its effectiveness in minimizing learning errors. Training involved 50 epochs with a batch size of 32, while early stopping was implemented to prevent overfitting during the training process.

The total count of parameters encompassing weights and biases across all model layers plays a role in determining the model’s ability to learn from data effectively. The settings were adjusted precisely to enhance the model’s ability to classify apnea incidents from ECG signals. The key elements and their respective setups in the CNN model are outlined in the model summary table showing the aspects utilized in this research.

#### 3.5.2. Dual-Branch Model

Figure 5 provides an overview of the layers in the dual-branch model, including their output shapes, parameter counts, and connections.

The dual-branch CNN model, designed with architecture and meticulous adjustments, proves its effectiveness as a tool for identifying sleep apnea occurrences from ECG signals. The CNN model outlined in the research utilizes an architecture with two branches for identifying apnea events based on ECG signals. These branches employ layers with different kernel sizes, followed by max pooling and batch normalization layers to extract and refine features from the input data. The extracted features are combined into a vector and processed through fully connected layers incorporating regularization techniques, such as L2 regularization and dropout, to enhance model generalization. The output layer employs a sigmoid function for the classification of apnea probability. Training involves a dataset with SMOTE for class imbalance, alongside stopping and learning rate adjustments for optimal performance tuning. Evaluation using validation and test sets demonstrated classification metrics, including ROC AUC scores, indicating the model’s proficiency in distinguishing between apnea and non-apnea events. The model includes two sets of layers, where each set utilizes different kernel sizes (three and five) to capture diverse patterns in the ECG data. These sets comprise max pooling layers and batch normalization layers. The results from both sets are combined into a feature vector, which is then passed through connected layers. To avoid overfitting, techniques such as L2 regularization and dropout are used. The final output layer employs a sigmoid activation function to estimate the likelihood of an apnea event in a classification scenario. The model underwent training, with a rounded dataset adjusting class weights to address any remaining imbalances. To prevent overfitting, early stopping was implemented along with modifications to the learning rate for convergence. Additionally, a system for checkpointing was utilized to maintain the performance of the model throughout the training process.

#### 3.5.3. Decision Tree Model

The machine learning model was used to identify sleep apnea based on ECG signals. Initially, the ECG signals were preprocessed by applying a bandpass filter to maintain frequencies at between 0.5 Hz and 45 Hz. These filtered signals were then divided into fixed 60 s segments. The model processed data from specified folders, splitting them into training and testing sets. The apnea annotations were converted into labels distinguishing apnea from non-apnea occurrences. The ECG signal segments were labeled accordingly. The decision tree classifier was then trained on the training set. The model’s performance was assessed on the training, validation, and test sets using metrics such as the classification report, confusion matrix, and ROC AUC scores. Finally, ROC curves and confusion matrices were generated to visualize how well the model differentiated apnea and non-apnea events.

### 3.6. Random Forest Classifier

The Random Forest model was developed to identify instances of sleep apnea in ECG data. This involved filtering, segmenting the ECG signals ensuring classes with SMOTE, and building a model with 100 decision trees. To assess the model’s effectiveness, ROC curves and confusion matrices were generated for the validation and test sets, indicating the accuracy and consistent detection of apnea events. The findings underscored the model’s ability to achieve accurate automated sleep apnea detection.

### 3.7. Performance Evaluation Metrics

The primary aim of the study was to categorize ECG signals as either apnea or non-apnea events. The performance evaluation metrics comprised accuracy, precision, recall, and F measure. Accuracy measures how close a predicted label is to the actual label, indicating the model’s capacity to accurately determine whether an ECG segment signifies an apnea event or a non-apnea event:(1)$$Accuracy=\frac{TP+TN}{TP+TN+FP+FN}$$

When it comes to ML and DL, precision is determined by dividing the number of instances by the total number of true positives and false positives. Recall, on the other hand, is calculated by dividing the number of positive instances by the overall count of actual positive instances in a specific category. The F measure, also known as the F1 score, is a measurement that merges precision and recall into a value that reflects their average. This metric is particularly valuable for evaluating how well a classification model performs, since it considers both precision and recall equally. The F measure varies between 0 and 1, with 0 indicating poor performance and 1 signifying optimal results. The formula for this metric can be expressed as follows (Equation (4)):(2)$$Precision=\frac{TP}{TP+FP}$$(3)$$Recall\;\mathrm{or}\;Sensitivity=\frac{TP}{TP+FN}$$(4)$$F\text{-}\mathrm{measure}=2\;*\;\frac{Precision*Recall}{Precision+Recall}$$

ROC curves are a tool for organizing and displaying how well classifiers perform. They are valuable when attempting to achieve a balance between identifying positives (sensitivity) and flagging negatives. ROC curves have been used for many years to gauge the reliability of tests. Nowadays, they are widely used in ML, DL, and data mining because they offer a way to visualize classifiers’ performance at different thresholds.

## 4. Experimental Results

### 4.1. CNN Model

The proposed convolutional neural network (CNN) model was highly effective at detecting apnea events from ECG signals based on classification. Model evaluation was consistent and the performance was robust from training to testing to validation, indicating good generalization capabilities. The model had exceptional accuracy, precision, recall, and F1 scores across all the datasets, indicating the high discriminative power of the model. More specifically, the training set classification report had nearly perfect precision, recall, and F1-scores, indicating that the model effectively learns and distinguishes between apnea and non-apnea events (refer to Table 2). The training confusion matrix also supports these findings, further supporting the low misclassification rate and showing that the model accurately differentiates classes for training. Precision and recall remain high in the validation set results, showing the model’s stability and efficacy beyond the training data. The validation confusion matrix also shows the high reliability of the model, with virtually no false positives or false negatives. Similarly, the test evaluation also confirms consistent performance, which indicates that the model is robust and does not have a significant propensity to overfit or bias itself. The test confusion matrices show low rates of misclassification, which supports the claim that the model maintains predictive reliability on unseen data (refer to Figure 5). The ROC curves confirm these findings by showing that the performance metrics are exceptional across all subsets. The ROC area under the curve (AUC) was almost a perfect shape—a triangle close to unity—suggesting excellent discriminative capability. This high ROC AUC metric indicates that the model has a high capacity to correctly distinguish apnea from normal respiratory states across different datasets, supporting its clinical relevance (see Figure 6).

To improve clinical trust and interpretability, model explainability methods such as integrated gradients and Grad-CAM were used. The integrated gradients analysis showed that ECG waveform components such as QRS complexes and waveform intervals were the most important factors in predicting apnea events. Grad-CAM gave visual proof by showing the areas of the ECG signals that were important for the convolutional layers and were therefore important features; these features were in sync with clinical understanding (see Figure 7). Thus, these results have high clinical relevance as they show that the model’s decisions are correlated with physiological patterns and clinical reasoning. Therefore, these results clearly support the validity of the model for reliable, clinically relevant apnea detection, making it a promising tool for assisting healthcare professionals in clinical practice.

The CNN ECG model’s training performance is consistently very high across all evaluation folds, indicating excellent fitting and effective data learning. The generalization to the unseen data is strong and the model’s training results are consistent with its validation performance. The test performance also further confirms this robust generalization, with only a slight reduction in accuracy compared to the training set, which is a normal and expected outcome. Therefore, these results clearly confirm the stability and appropriateness of the CNN model for ECG classification tasks (refer to Table 3). The CNN model was evaluated using multiple iterations within X-fold cross-validation, achieving an average ROC AUC of 0.9487 ± 0.0274 and an accuracy of 0.8783 ± 0.0378, demonstrating robust performance consistency. These results show that the CNN model maintains high ROC AUC and accuracy on multiple cross-validation iterations, indicating its reliability and effectiveness in detecting sleep apnea from ECG signals.

### 4.2. Dual-Branch Model

The developed model, a dual-branch convolutional neural network (CNN) model for apnea detection using ECG signals, showed high performance in all training, validation, and test datasets. The classification metrics of the model reveal very high accuracy, precision, recall, and F1 scores, indicating the ability of the model to distinguish between apnea and non-apnea segments robustly. The results show that the training set had near-perfect precision and recall, which shows that the dual-branch architecture could learn the patterns associated with apnea in the ECG signals. The validation set had accurate, specific, and sensitive values, which shows that the model is likely to perform well generally and not just with the training data. The performance of the test dataset reinforced these findings, with the consistently high precision, recall, and F1 scores indicating that the model is robust and dependable when used on new data (see Table 4).

ROC curve analysis provided additional insights into the discriminative power of the dual-branch CNN model. The ROC curve (AUC) reaches unity across training, validation, and test sets, which indicates that the model is very good at distinguishing between apnea events and normal breathing patterns. The ROC AUC values remain consistently high, which supports the model’s clinical readiness as it can reliably identify apnea events from ECG signals with a low rate of both false positives and false negatives (refer to Figure 8).

Quantitative metrics are supported by confusion matrices for the training, validation, and test datasets, with visually minimal misclassification instances observed. The training confusion matrix shows almost no misclassification, which is encouraging given the excellent learning capacity of the model. The validation confusion matrix results show a slight increase in the rate of misclassifications, but these remain minimal and support the strong generalization capabilities of the model. The results of the test confusion matrix are also good, showing that the model is robust and has a low error rate when identifying apnea events from ECG signals in different datasets (see Figure 9).

Table 5 shows that the dual-branch CNN model has excellent performance; it shows high accuracy (0.896 ± 0.138) and a good ROC AUC (0.9019 ± 0.0271). Overall, the model can balance sensitivity and specificity and can detect apnea from ECG signals. These results confirm the reliability and correctness of the model in detecting apnea events from ECG signals and the stability of the generalization capability across multiple cross-validation runs. This also shows that the proposed CNN architecture is suitable for clinical apnea detection applications and that there is a significant improvement in accuracy and consistency over previous methods.

Grad-CAM visualization was employed to further validate the ability to interpret the model. Grad-CAM clearly identified the regions of the ECG that influence the model’s prediction most strongly, highlighting key morphological features such as the QRS complexes and specific intervals that are critical for apnea detection. The brightly colored areas of the Grad-CAM visualization indicate the segments the convolutional layers emphasized, providing crucial insight into the model’s decision-making process. The model’s capacity to focus accurately on clinically relevant waveform features greatly enhances its interpretability and clinical trustworthiness, such that clinicians can have confidence in the underlying rationale of the predictions made (refer to Figure 10 and Figure 11).

### 4.3. Random Forest

The RF model performed well during validation, achieving a score of 1.00 for precision, recall, and F1 scores in both the non-apnea (0) and apnea (1) categories. It accurately identified all 10,220 samples in the validation set with 100% accuracy. The confusion matrix revealed the classification of apnea cases (5070 correctly identified) and the almost perfect classification of apnea cases with only four misclassifications. The ROC AUC score was 0.9999997702073877, indicating the model’s excellent performance in distinguishing between apnea and non-apnea occurrences. However, during testing, the model exhibited a precision of 0.87 in apnea (0) scenarios, correctly predicting non-apnea cases 87% of the time. The recall rate for non-apnea was at 0.64, signifying that it identified 64% of apnea instances accurately. The F1 score for non-apnea was 0.74, representing a blend of precision and recall measures. Conversely, the model achieved high precision, recall, and F1 scores (1.00) for apnea (1), showing high performance in detecting apnea events. The overall accuracy on the test set reached 100%, demonstrating the model’s classification across samples. The confusion matrix indicated that the model accurately identified most apnea cases (17,236) and 52 non-apnea cases. However, it mistakenly categorized 29 non-apnea instances as apnea and 8 apnea instances as non-apnea. The ROC AUC score for the test set was 0.98, showing less-than-perfect performance compared with the validation set (refer to Figure 12). The RF model performed well in the validation phase, achieving a 100% accuracy rate and a perfect ROC AUC. This shows that the model successfully learned to differentiate between apnea and non-apnea events. The model’s ability to accurately detect apnea was further demonstrated by its precision, recall, and F1 scores for apnea detection in both the validation and test datasets. This level of performance underscores the model’s reliability in identifying apnea cases, which is crucial in diagnostics where missing a true apnea case could lead to serious consequences. However, the model encountered difficulties when it came to detecting apnea cases, particularly during the test phase.

The precision and recall of non-apnea cases were lower, with a recall of 0.64 indicating that the model missed several non-apnea cases (refer to Table 6). This might suggest a tendency towards predicting apnea due to an imbalance in the class distribution within the data. Despite these challenges, the model still maintained some level of accuracy, indicating its ability to generalize well to new data; however, there is room for improvement when it comes to handling non-apnea cases.

The RF model is quite dependable when it comes to spotting apnea episodes, as shown by the identified cases; however, it faces challenges in categorizing apnea instances, with a notable number of false positives (mislabeling non-apnea cases as apnea). This indicates that, although the RF model excels at detecting apnea, there might be room for refinement or incorporating features to enhance its ability to differentiate apnea events (see Figure 13).

### 4.4. Decision Tree

The decision tree model showed good performance in both the training and validation phases; however, it faced challenges during testing. During training, the model achieved high precision, recall, and F1 scores for both apnea and non-apnea classes, accurately classifying all 23,846 samples without any errors. This resulted in a 100% accuracy rate and a ROC AUC score of 1.0, showing high performance in distinguishing between apnea and non-apnea events (see Figure 14). During validation, the model maintained precision, recall, and F1 scores for both classes that were similar to the training performance. The accuracy remained at 100%, successfully classifying all 10,220 samples. Twelve apnea cases were misclassified according to the confusion matrix, while the ROC AUC score slightly decreased to 0.9988 but still indicated high performance in distinguishing between the two classes. However, there were differences in the model’s performance on the test set, especially in detecting apnea cases, where the precision dropped to 0.53, indicating correct predictions only 53% of the time for non-apnea cases. The non-apnea recall rate was 0.58, showing that the model accurately identified 58% of non-apnea cases, resulting in an F1 score of 0.56 (see Table 7). Despite this, the model excelled in detecting apnea cases, with precision, recall, and F1 scores of 1.00. The overall accuracy of the test set remained high at 100%, mainly due to classifying several apnea cases. The confusion matrix revealed that, while most apnea cases were accurately classified, there were challenges with non-apnea cases, with 34 instances mistakenly categorized as apnea and 41 apnea instances categorized as non-apnea. The ROC AUC score decreased to 0.7889 on the test set, indicating a decrease in performance in distinguishing between apnea and non-apnea events compared to the training and validation phases.

The results of the decision tree model show a level of accuracy when classifying apnea events during the training and validation phases, where it did exceptionally well. However, there is a decrease in performance on the test set in identifying non-apnea cases, which is a cause of some concern. The good performance during training and validation indicates that the model has learned the patterns within those datasets effectively. It could also suggest overfitting. Overfitting happens when a model excels with the training data but struggles to adapt to the test data. The decrease in performance on the test set and the reduced precision and recall for apnea cases support this idea. While the model’s ability to detect apnea cases is promising, in medical scenarios where accurate identification is essential, its lower accuracy in recognizing non-apnea cases may result in false positives. This could lead to non-apnea instances being wrongly classified as apnea events, potentially causing unnecessary stress for patients or unwarranted medical interventions.

The DT model performed excellently during training and validation but faced challenges with the test set, especially in cases without apnea. This difference indicates that, although the model works well in certain situations, its capacity to adapt to varied data is limited, perhaps because it is too tailored during training (see Figure 15).

As shown in Figure 16, the authors of [30] presented an approach for identifying sleep apnea (SLA) by analyzing single-lead ECG signals. It uses a deep belief network (E DBN) incorporating two types of restricted Boltzmann machines (RBMs). After being trained on the apnea ECG dataset, the model achieved an accuracy of 89.11% per segment and 97.17% per recording, offering an affordable option compared with polysomnography tests. The authors of [31] identified sleep apnea by analyzing ECG signals using a neural network (DNN) that incorporates LSTM and BiLSTM models. The BiLSTM model, which utilizes backward learning, attained a high accuracy rate on the Physionet dataset, showing exceptional capacity to detect sleep apnea.

The authors analyzed how the structure of learning models and the length of label mappings impact personalized transfer learning (TL) for detecting sleep apnea using ECG data. The hybrid transformer model (HTM) demonstrated better performance over a CNN model, achieving an accuracy of 85.37% and an AUC of 0.9147 compared with CNN’s accuracy of 84.12% and AUC of 0.9002. The study highlights that increasing the length of label mappings enhances model performance, with HTM excelling in both general and personalized TL scenarios. Positive samples contribute to improving TL within the database, while negative samples enhance TL across databases [32]. In [33], the authors present a deep learning approach using a modified Fusion Convolutional Neural Network (MFCNN) to analyze sleep apnea’s impact on cancer progression. The model evaluates variables such as throat muscle collapse during sleep, which causes snoring and gasping, and investigates the relationship between low oxygen levels, sleep apnea, and cancer impermanence. The study highlights the MFCNN’s ability to detect sleep apnea occurrences in ECG data and explores the complexities linking sleep apnea with cancer development [33]. In [34], DL was used to classify sleep apnea–hypopnea severity using a single airflow (AF) signal, simplifying the traditional PSG method. The DL model achieved 83.46%, 85.39%, and 92.69% accuracy in binary classification and 63.70% in multiclass classification, outperforming SVM and Adaboost-CART on 520 records from the MrOS sleep study. The authors of [35] showed that signs of OSA can be detected automatically from single-lead ECG signals using a 1D Convolutional Neural Network (1DCNN). The 1DCNN performed better than the SVM and Random Forest models, reaching an accuracy of 96.99% along with a sensitivity of 0.9743 and specificity of 0.9708. This presents a method for OSA diagnosis. The authors of [35] used supervised machine learning to analyze polygraphy data for sleep apnea diagnosis, achieving up to 89.41% accuracy. Deep learning methods significantly outperformed the other classifiers, with convolutional neural networks excelling in both performance and efficiency. In [36], the researchers used advanced techniques, such as deep learning and machine learning, to detect sleep apnea using ECG data. The study demonstrated an 88.13% accuracy rate in identifying the condition using a five cross-validation approach, showing the reliability of these methodologies in providing precise diagnoses. Recent research [37] suggests an approach that blends deep learning models such as DNN, GRU, RNN, and LSTM with a machine learning meta-learner to identify sleep apnea. It showed 95.74% accuracy in detecting apnea and 99.4% in classifying apnea types. This method stands out for its precision and reliability, helping to diagnose sleep disorders at facilities.

In contrast, our state-of-the-art study emphasizes the advantages of using learning to detect sleep apnea from ECG data. The E DBN model [31] achieved high accuracy (97.17% per recording) by utilizing restricted Boltzmann machines, while the BiLSTM model [32] demonstrated high performance on the Physionet dataset. Although our CNN model achieved high accuracy, it encountered difficulties in apnea detection, similar to the challenges observed in other models, such as the hybrid transformer model (HTM) [34]. The dual-branch CNN in our study closely corresponds with the precision demonstrated using the MFCNN approach [35], which associated sleep apnea with cancer progression. The various models investigated in these studies, including those mentioned in [36,37], underscore the growing efficacy of learning in this field. Our Dual-Branch CNN model offers a tool for diagnosing sleep apnea that is superior to the available advanced approaches.

For OSA detection from single-lead ECG signals, a deep learning model based on EfficientNet is used together with XGBoost. The model converts raw ECG signals to spectrogram images using a technique known as short-time Fourier transform (STFT) to capture time–frequency features easily. Overlapping slicing guarantees that the model has a higher chance of capturing complete apnea events, thus increasing the sensitivity of the detection. Using sample weight adjustments during training is a good way of managing class imbalances. An integrated deep neural network is proposed for directly classifying the raw EEG signals into different levels of OSA severity. This integrated approach achieves high accuracy (ACC = 0.928, AUC = 0.975) in identifying OSA severity, reducing computational complexity and presenting a clinically viable, cost-effective alternative to traditional polysomnography, thus improving reliable patient screening.

Sheta et al. (2021) proposed a diagnostic framework for obstructive sleep apnea (OSA) using electrocardiogram (ECG) signals obtained from the PhysioNet Apnea ECG Database [18]. They extracted nine features from ECG signals after conducting initial preprocessing using a noise removal method. For classification, they compared the performance of thirteen machine learning and four deep learning algorithms and found that the bidirectional long short-term memory network outperformed the others with an accuracy of 92.18%. The results showed high robustness in diagnostic capability, proving the effectiveness of using machine learning in clinical settings. The advantage of this technique is that it offers significant clinical utility by allowing for the reliable, automated detection of OSA using a non-invasive, cost-effective ECG-based system, thereby reducing reliance on conventional polysomnography. For non-apnea cases, a precision of 0.58 means a high false-positive rate that will lead to unnecessary medical interventions, more burden on the healthcare system, patient anxiety, and distrust in AI-based diagnosis. The model may fail to accurately distinguish between different conditions, which can lead to overdiagnosis, with subsequent requirements for additional follow-up tests and the inappropriate allocation of resources, thus calling into question the model’s clinical reliability.

## 5. Conclusions

The current study establishes the possibility and efficacy of using sophisticated DL techniques such as CNNs and dual-branch CNNs for identifying sleep apnea from ECG signals. These techniques are superior to traditional methods, such as DT and RF, in terms of accuracy and precision. The dual-branch CNN was found to have a greater ability to detect the complicated ECG signals associated with apnea events, thus validating the effectiveness of such specialized neural network architectures for health informatics. However, both DT and RF models have drawbacks. They are likely to overfit the data and are not very accurate when applied to non-apnea instances. Future work should address the model’s limitations, for example, by using diverse datasets, improving the handling of class imbalance, and investigating hybrid modeling strategies to improve diagnostic accuracy.

## Figures and Tables

**Figure 1 biomedicines-13-01090-f001:**
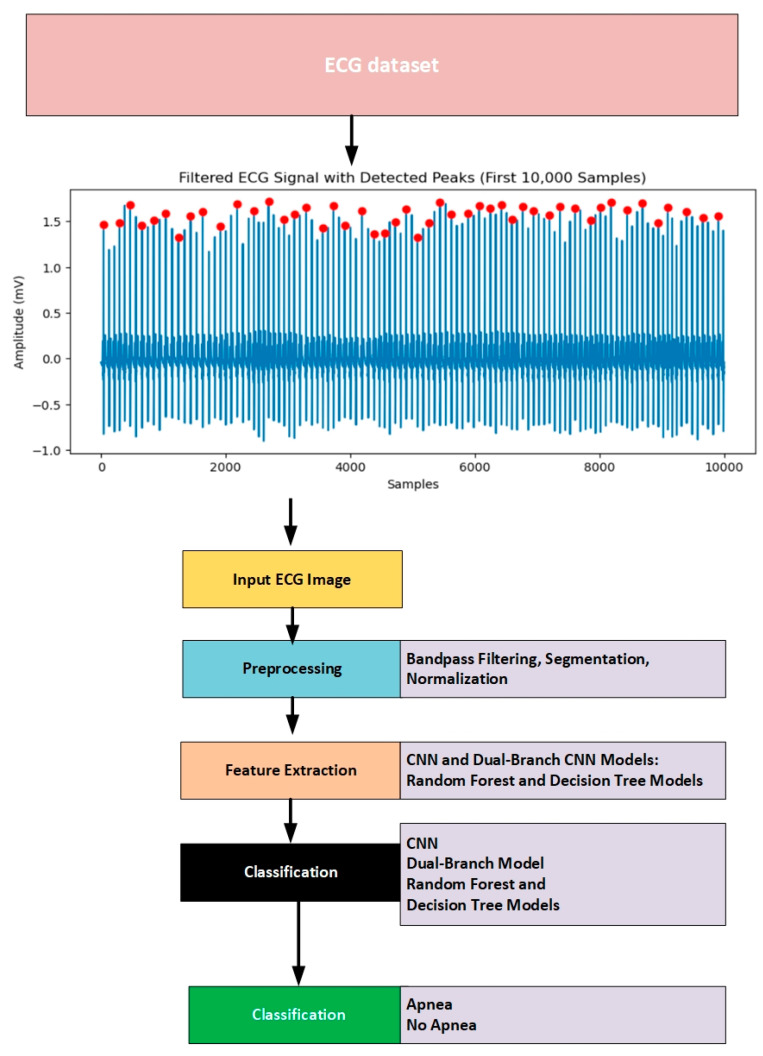
ECG-based apnea detection using dual-branch convolutional deep learning methods and machine learning techniques.

**Figure 2 biomedicines-13-01090-f002:**
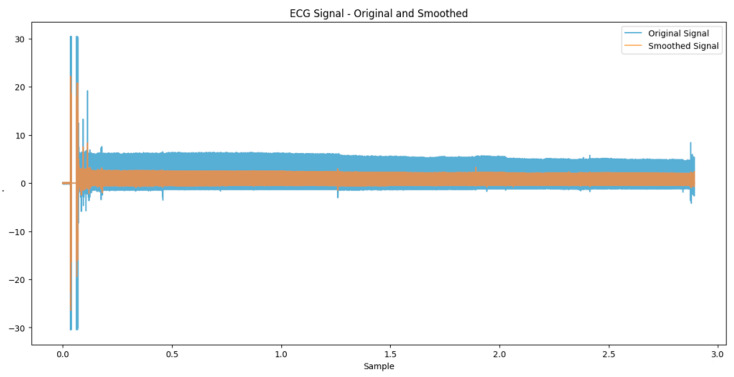
Comparison of original and smoothed ECG signals in the apnea dataset.

**Figure 3 biomedicines-13-01090-f003:**
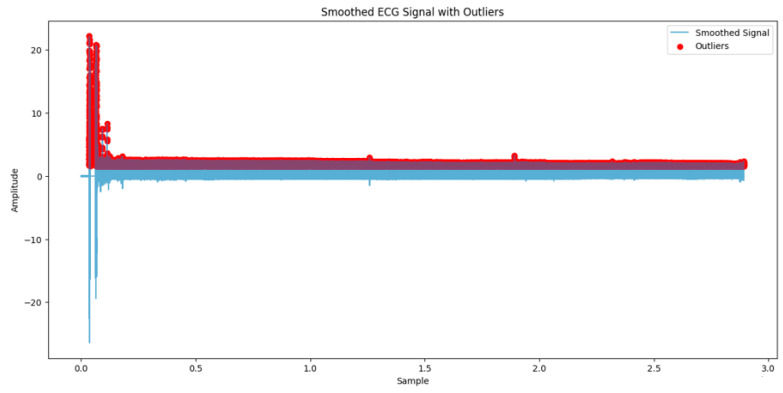
Smoothed ECG signal with detected outliers.

**Figure 4 biomedicines-13-01090-f004:**
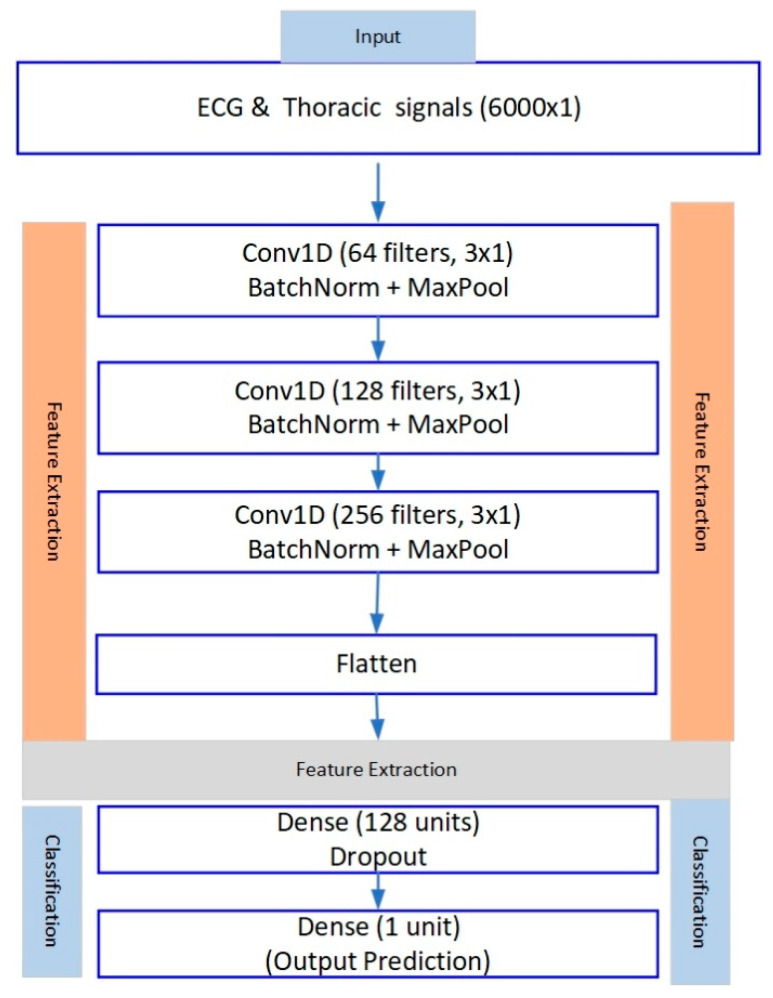
Convolutional neural network (CNN) model architecture.

**Figure 5 biomedicines-13-01090-f005:**
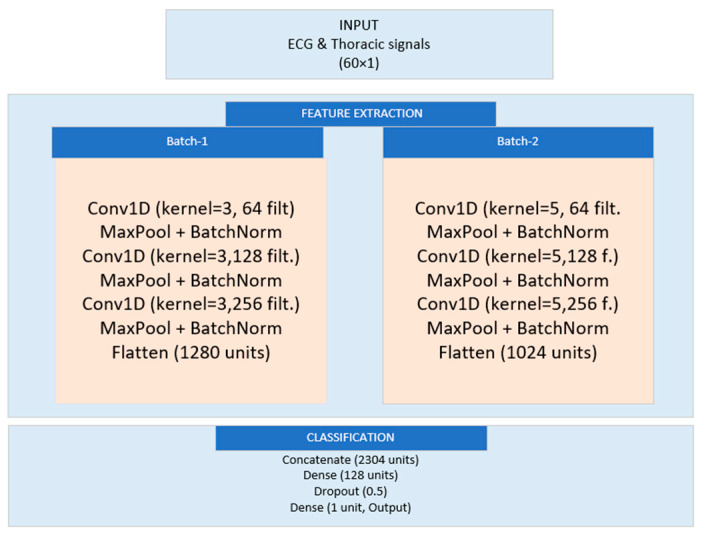
Summary of the dual-branch model architecture.

**Figure 6 biomedicines-13-01090-f006:**
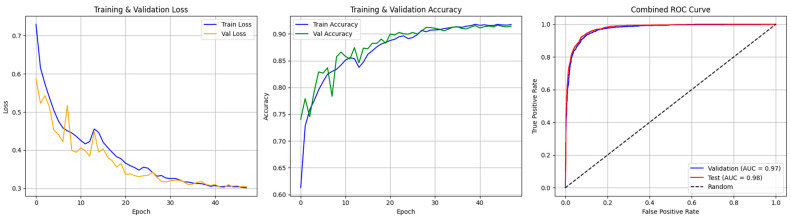
Accuracy and loss and ROC AUC graph of the test and validation of CNN model.

**Figure 7 biomedicines-13-01090-f007:**
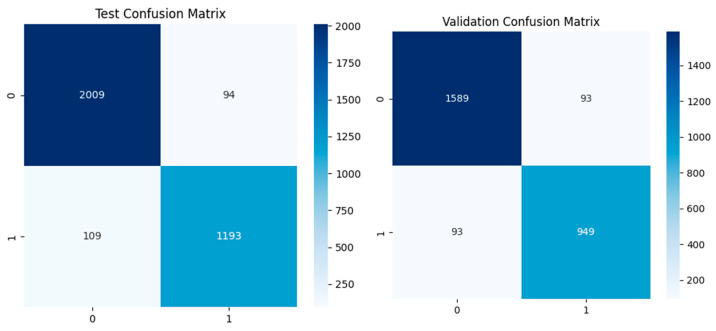
Confusion matrix of the CNN model for test and validation.

**Figure 8 biomedicines-13-01090-f008:**
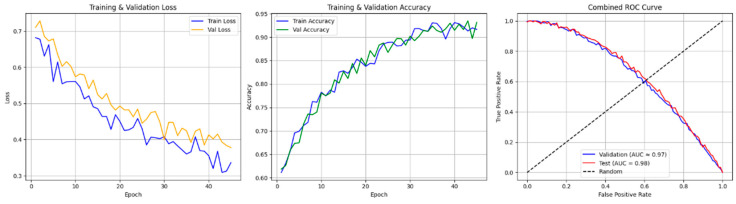
Accuracy and loss and ROC AUC graph of the testing and validation of the dual-branch model.

**Figure 9 biomedicines-13-01090-f009:**
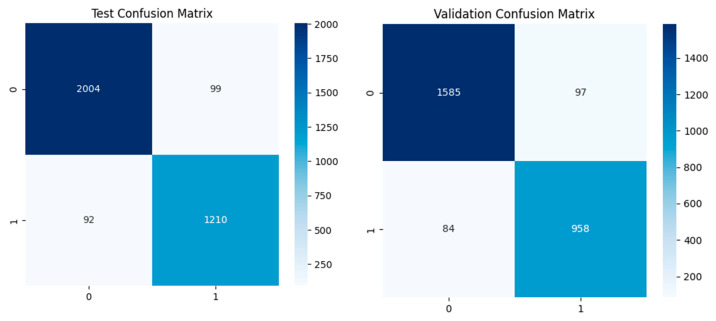
Confusion matrices of the dual-branch model for the training, testing, and validation sets.

**Figure 10 biomedicines-13-01090-f010:**
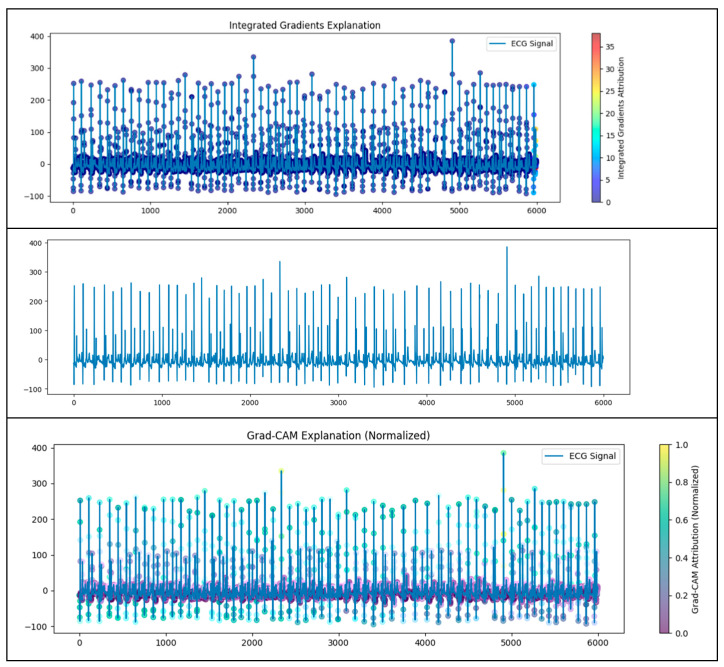
Grad-CAM and integrated gradient.

**Figure 11 biomedicines-13-01090-f011:**
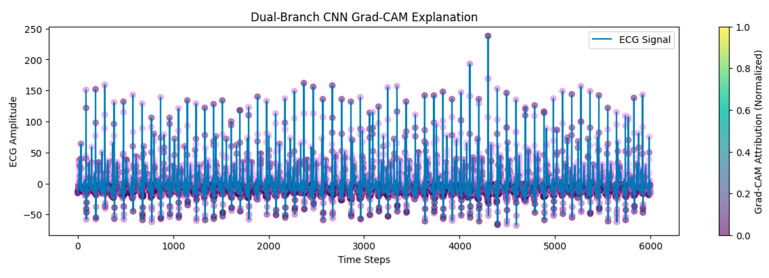
Grad-CAM and gradient.

**Figure 12 biomedicines-13-01090-f012:**
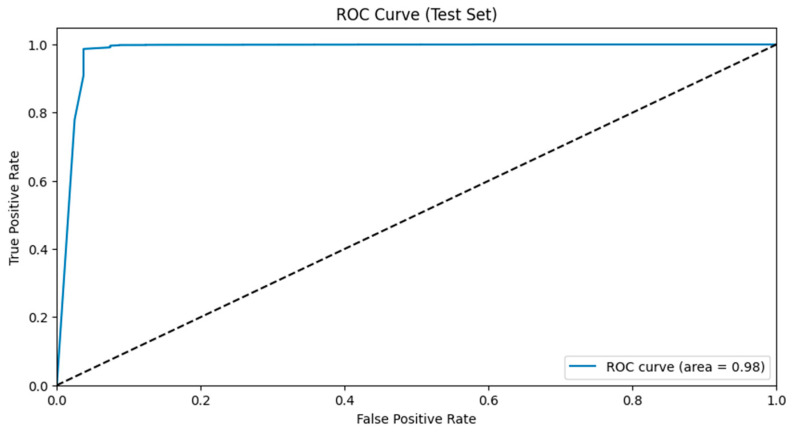
ROC AUC graph of the test set analyzed using the RF model.

**Figure 13 biomedicines-13-01090-f013:**
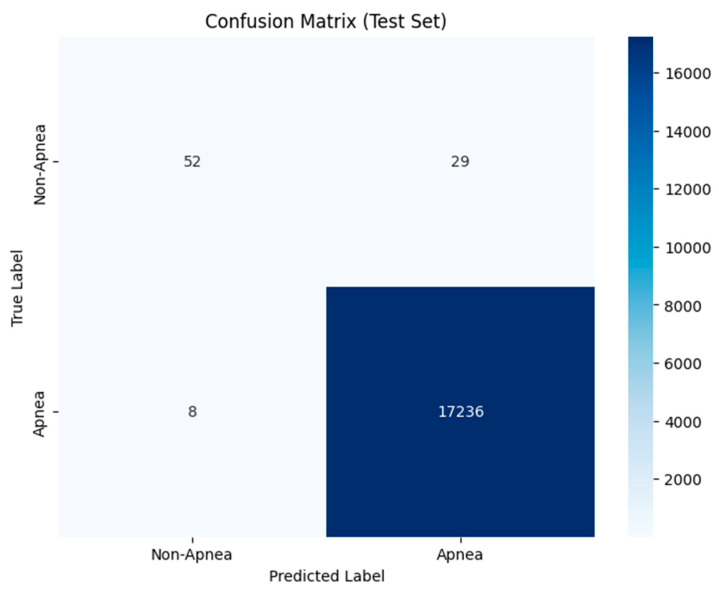
Confusion matrix of the test set analyzed using the RF model.

**Figure 14 biomedicines-13-01090-f014:**
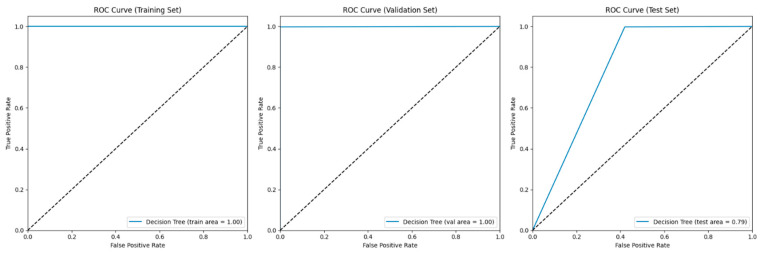
ROC AUC graphs of the testing, validation, and training of the RF model.

**Figure 15 biomedicines-13-01090-f015:**
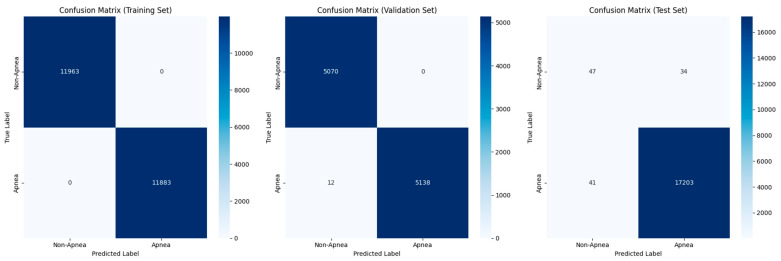
Confusion matrix of the DT model for the training, validation, and test sets.

**Figure 16 biomedicines-13-01090-f016:**
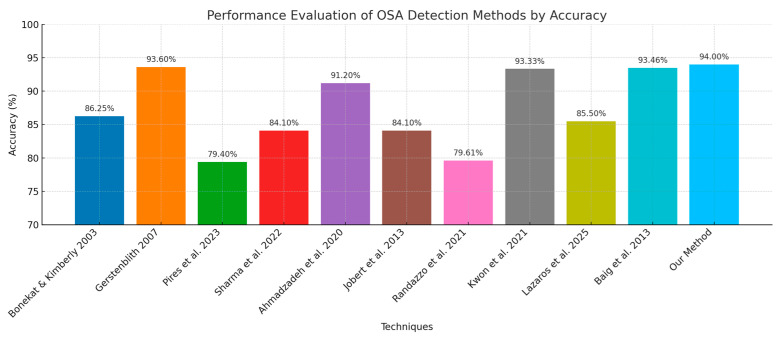
Comparison of our model with current studies [1,2,4,5,6,7,8,9,10,11].

**Table 1 biomedicines-13-01090-t001:** Summary of the state-of-the-art techniques, accomplishments, and issues associated with obstructive sleep apnea (OSA) detection.

Reference	Technique Used	Achievements	Problems Identified
[18] Sheta et al. (2021)	Dual-branch CNN on ECG signals	Higher accuracy	No significant drawbacks mentioned
[19] Bernardini et al. (2021)	AIOSA deep learning model (heart and breathing signals)	Higher accuracy	Computationally expensive; sensitive to signal noise; poor explainability
[20] Kim et al. (2021)	KL-6 biomarker blood test	Identified OSA severity	Invasive method
[21] Huang et al. (2024)	Logistic regression with BMI and TyG index	Low AUC	Limited validation
[22] Li et al. (2024)	ResNeSt34 deep learning (ECG + thoracic signals)	High Accuracy	Works well with time series text data but reduced performance when used with ECG
[23] Yue et al. (2021)	Multi-resolution residual network (Mr-ResNet) with nasal airflow	Excellent accuracy and sensitivity	Invasive method
[24] Monna et al. (2022)	3D maxillofacial shape analysis and machine learning	High accuracy	3D Scanning is needed, meaning extra load on the CPU
[25] Cen et al. (2018)	CNN	Real-time OSA event detection accuracy ~79.61%	Complex; has low performance when signals are mixed
[26] Nassehi et al. (2024)	KNN classifier on resting-state EEG	High accuracy (93.33%), excellent AUC (0.98)	Dataset is very small
[27] Liu et al. (2024)	EfficientNet + XGBoost on single-lead ECG	Excellent AUC (0.917–0.975), high accuracy (85.5–92.8%)	Model interpretation focuses only on specific apnea events, meaning that the model has difficulty identifying hypopneas with arousals
[28] Jiménez-García et al. (2024)	CNN + RNN on airflow and oximetry signals (pediatric)	High diagnostic accuracy (≥84%), good interpretability using Grad-CAM	Only for pediatrics

**Table 2 biomedicines-13-01090-t002:** Classification report of CNN model under test and validation evaluation.

Dataset	Class	Precision	Recall	F1-Score
Validation	0	0.94	0.94	0.94
	1	0.91	0.91	0.91
Test	0	0.96	0.95	0.95
	1	0.93	0.92	0.92

**Table 3 biomedicines-13-01090-t003:** Performance metrics (mean ± STD) across five folds of the CNN model.

Fold	Accuracy	ROC AUC
Fold 1	0.8821	0.9542
Fold 2	0.8614	0.9426
Fold 3	0.9137	0.9761
Fold 4	0.8513	0.9274
Fold 5	0.8829	0.9432
Mean ± SD	0.8783 ± 0.0378	0.9487 ± 0.0274

**Table 4 biomedicines-13-01090-t004:** Classification report of the dual-branch model for the training, testing, and validation tests.

Dataset	Class	Precision	Recall	F1-Score
Validation	Non-apnea (0)	0.95	0.94	0.95
	Apnea (1)	0.91	0.92	0.91
Test	Non-apnea (0)	0.96	0.95	0.95
	Apnea (1)	0.92	0.93	0.93

**Table 5 biomedicines-13-01090-t005:** Performance metrics (mean ± STD) across five folds of the dual-branch model.

Fold	Accuracy	ROC AUC
Fold 1	0.882	0.9291
Fold 2	0.857	0.8967
Fold 3	0.945	0.9175
Fold 4	0.721	0.8710
Fold 5	0.875	0.8951
Mean ± SD	0.896 ± 0.138	0.9019 ± 0.0271

**Table 6 biomedicines-13-01090-t006:** Classification report of the RF model under test and validation evaluation tests.

Random Forest Model Validation			
Apnea	1	1	1
Non-apnea	1	1	1
**Random Forest Model Test**			
Apnea	0.87	0.64	0.74
Non-apnea	1	1	1

**Table 7 biomedicines-13-01090-t007:** Classification report of the DT model under test and validation evaluation tests.

DT Model Validation			
Apnea	1	1	1
Non-apnea	1	1	1
**DT Model Test**			
Apnea	0.53	0.58	0.56
Non-apnea	1	1	1

## Data Availability

The datasets generated and/or analyzed during the current study are available in the PhysioNet repository at https://physionet.org/content/apnea-ecg/1.0.0/, accessed on 20 April 2024.

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
