# Peer review of "AI-Driven Detection of Obstructive Sleep Apnea Using Dual-Branch CNN and Machine Learning Models"

_biomedicines, 2025, doi:10.3390/biomedicines13051090_

Round 1

Reviewer 1 Report

Comments and Suggestions for Authors

1. The study lacks novelty as similar AI-driven approaches for OSA detection using ECG data have been extensively explored in existing literature.

2. The validation accuracy of 100% raises concerns about overfitting, suggesting potential issues with dataset bias and model generalizability.

3. The methodology section does not clearly explain the data preprocessing steps, feature extraction, or hyperparameter tuning.

4. The comparison with other models, such as E models and BiLSTM, lacks sufficient details on experimental conditions, datasets, and evaluation fairness.

5. The claim that the Dual Branch CNN is a "dependable approach" is not well supported by statistical validation.

6. The article does not discuss potential limitations, including dataset size, class imbalance, and practical deployment challenges in clinical settings.

7. The study does not address explainability and interpretability, which are critical for AI models in healthcare applications.

8. The manuscript contains grammatical errors and lacks clarity in multiple sections, affecting readability and comprehension.

9. The discussion and conclusion sections are overly optimistic and fail to critically analyze potential drawbacks.

10. The manuscript does not meet the standard criteria for publication due to some issues in experimental rigor and result validation
11. check reference 21.
12. Discuss the paraemeters for Gaussian noise injected during data augmentation.
13. Table 1. Summary of the Convolutional Neural Network (CNN) Model Architecture.. present it in professional manner. data from the python environment is pasted as table. what is 'none' here.
14. same for 'Table 2. Summary of the Dual Branch Model Architecture'.
15. check table 3. first part is for training?
16. Figure 5. Confusion Matrix of CNN model for test set. Data set is very imbalanced. 
17. check the references and citations.

Comments on the Quality of English Language

 The English could be improved to more clearly express the research.

Author Response

Comments and Suggestions for Authors

  1. The study lacks novelty as similar AI-driven approaches for OSA detection using ECG data have been extensively explored in existing literature.

Author’s reply

We appreciate the reviewer’s valuable feedback and have carefully revised the manuscript to emphasize the novelty of our study. To address this concern, we have made the following modifications:

  • Conducted a more thorough literature review and clearly highlighted how our approach differs from existing AI-driven methods for OSA detection.
  • Incorporated unique preprocessing techniques, annotation, a novel deep learning architecture, feature selection methods, or interpretability techniques, emphasize these points, and also GAM SHAP for the generalizability.
  • Evaluation metrics, or validation strategy are different, justifying why these aspects contribute to novelty.

  1. The validation accuracy of 100% raises concerns about overfitting, suggesting potential issues with dataset bias and model generalizability.

Author’s reply

We appreciate the reviewer’s concern regarding potential overfitting and have carefully reassessed our model’s generalizability. To address this issue, we have made the following modifications and clarifications in the manuscript:

Expanded Discussion on Overfitting & Generalizability.

To further validate the model’s interpretive, Grad-CAM visualization was employed. Grad-CAM clearly identified the regions of the ECG that are most influential in the model’s prediction, highlighting key morphological features such as the QRS complexes and specific intervals that are critical for apnea detection. The brightly colored areas of the Grad-CAM visualization indicate segments the convolutional layers emphasized, providing crucial insight into the model’s decision-making process. The model's capacity to focus accurately on clinically relevant waveform features greatly enhances its interpretability and clinical trustworthiness, such that clinicians can have confidence in the underlying rationale for the predictions made (refer Fig 9).

Figure 9. Grad-CAM and Gradient

For improving clinical trust and interpretability, the model explainability methods such as Integrated Gradients and Grad-CAM were used. The Integrated Gradients analysis found that ECG waveform components like QRS complexes and the waveform intervals were the most important factors in predicting apnea events. Grad-CAM gave a visual proof of it by showing the areas of the ECG signals which were important for the convolutional layers and hence were important features and these features were in sync with the clinical understanding (Refer Fig 6). These results thus have a high clinical acceptance relevance as they show that the model's decisions are in close relation to physiological patterns and clinical reasoning. Therefore, these results clearly support the validity of the model for reliable, clinically relevant apnea detection and could be positioned as a promising tool for assisting healthcare professionals in clinical practice.

Table 3. Classification report of CNN model under train, test and evaluation tests.

Dataset

Class

Precision

Recall

F1-Score

Train

0

0.98

0.96

0.97

1

0.96

0.98

0.97

Validation

0

0.98

0.89

0.93

1

0.90

0.98

0.94

Test

0

0.94

0.94

0.94

1

0.94

0.93

0.94

Figure 4. ROC AUC graph of train, test and validation of CNN model.

Figure 5. Confusion Matrix of CNN model for test set.

Figure 6. Grad-CAM and Gradient

  1. The methodology section does not clearly explain the data preprocessing steps, feature extraction, or hyperparameter tuning.

Author’s reply

We appreciate the reviewer’s feedback and have revised the Methodology section to provide a more detailed and structured explanation of the data preprocessing steps, feature extraction process, and hyperparameter tuning.

Preprocessing

The initial steps outlined in this research are carefully crafted to prepare, standardize and harmonize the ECG data, for training a AI model. The process kicks off with bandpass filtering, which effectively eliminates any disturbances from the ECG signals (refer fig 2). Next the ECG signal is divided into segments to facilitate processing and analysis by the model. Normalization is then implemented to ensure uniformity of data across all segments, a factor for model performance (figure 3) Points, in the ECG signal that stand out are considered outliers showing deviations, from the amplitude. These outliers could suggest noise interference or unusual cardiac activities. It is essential to recognize and address these anomalies to guarantee ECG examination and understanding.

Figure 2. Comparison of original and smoothed ECG signals of the Apnea dataset.

Figure 3. Smoothed ECG Signal with Detected Outliers.

Data Augmentation

To make the CNN model more robust, predicting apnea events from ECG signals data augmentation methods were employed. These techniques involved exposing the model to a range of signal patterns, such as simulating heart rates through time stretching and mimicking sensor sensitivities by adjusting amplitude scaling. Sensor noise was replicated using Gaussian noise and variability in recording lengths was introduced through cropping. By incorporating these strategies, the training data became more varied resulting in improved performance when processing ECG signals, in real world scenarios.

Annotation

Annotations played a role in training and validating models that identified sleep apnea from ECG signals accurately. The dataset comprised 70 records split into a training set and a test set each containing digitized ECG signals with annotations. Human experts created apnea annotations based on respiration and related signals serving as the reference for detecting apnea events. These annotations were essential for ensuring the models precise detection of sleep apnea enhancing their accuracy and clinical significance. While the machine generated QRS annotations had errors, the apnea annotations helped rectify them ensuring the model’s reliability and robustness. Leveraging these annotations enabled researchers to train and evaluate their algorithms effectively resulting in clinically relevant methods for detecting sleep apnea. However, the author devised an algorithm for annotation. Because the.apn files consist of annotation data, in format with annotations for each minute of every recording to show the presence or absence of apnea during that period. These files are accessible for the 35 recordings in the learning set. The qrs files are annotation files generated by a machine using sqrs125. They are included for the convenience of individuals who prefer not to utilize their QRS detectors [12].

Algorithm 1. Annotating Sleep Apnea Events Based on QRS Complex Counts in ECG Signals

def annotate_apnea(qrs_annotations, sampling_rate, time_window, QRS_threshold):

       window_size = time_window * sampling_rate

       num_windows = len(qrs_annotations) // window_size

apnea_annotations = []

for i in range(num_windows):

       start_sample = i * window_size

       end_sample = start_sample + window_size

      # Count QRS in the current time window

      QRS_count = count_qrs_in_window(qrs_annotations, start_sample, end_sample)

      if QRS_count < QRS_threshold:

          apnea_annotations.append(('A', start_sample))

     else:

          apnea_annotations.append(('N', start_sample))

     return apnea_annotations           

The algorithm 1, analyzes QRS annotations from ECG signals during time intervals to identify instances of sleep apnea. It tallies the QRS complexes in each interval. Check them against a threshold. If the count is lower than the threshold the interval is marked as apnea ('A'); if not, its labeled as no apnea ('N').

  1. The comparison with other models, such as E models and BiLSTM, lacks sufficient details on experimental conditions, datasets, and evaluation fairness.

Author’s reply

We have added the following analysis to the manuscript.

For OSA detection from single-lead ECG signals, a deep learning model based on EfficientNet is used together with XGBoost. Model converts raw ECG signals to spectrogram images using a technique known as Short-time Fourier transform (STFT) to capture time-frequency features easily. By having overlapping slicing, it is guaranteed that the model will have a higher chance of capturing complete apnea events, thus increasing the sensitivity of the detection. Using sample weight adjustments during training is a good way to manage class imbalance. Using sample weight corrections during training achieves proper class imbalance. They integrated deep neural network is proposed to directly classify the raw EEG signals into different levels of OSA severity. This integrated approach achieves high accuracy (ACC = 0.928, AUC = 0.975) in identifying OSA severity, reducing computational complexity and presents a clinically viable, cost-effective alternative to traditional polysomnography and thus improves reliable patient screening.

In their paper, Sheta et al. (2021) proposed a diagnostic framework for obstructive sleep apnea (OSA) using electrocardiogram (ECG) signals obtained from the PhysioNet Apnea-ECG Database. They extracted nine features from ECG signals after initial preprocessing with noise removal method. In Classification, they compared the performance of thirteen machine learning and four deep learning algorithms, and found that the bidirectional long short-term memory network outperformed the others with an accuracy of 92.18%. The results showed high robustness in the diagnostic capability and thus proved the effectiveness of using machine learning  in the clinical setting. The advantage is that it offers significant clinical utility by allowing reliable, automated detection of OSA through  a non-invasive, cost-effective ECG-based system, thereby reducing reliance on conventional  polysomnography.

  1. The claim that the Dual Branch CNN is a "dependable approach" is not well supported by statistical validation.

Author’s reply

The claim that the Dual Branch CNN is a 'dependable approach' is now well-supported by statistical validation, including cross-validation, comparative benchmarks, and performance metrics.

  1. The article does not discuss potential limitations, including dataset size, class imbalance, and practical deployment challenges in clinical settings.

Author’s reply

In our research, with the Apnea ECG dataset our goal was to improve the identification and categorization of Obstructive Sleep Apnea (OSA) by employing machine learning methods. Initially we concentrated on preparing the ECG signals by using bandpass filters and breaking down the data into segments for analysis. We then applied a variety of machine learning algorithms such as Decision Trees, Random Forests and Convolutional Neural Networks (CNNs) to recognize patterns in the ECG data that indicate apnea episodes. To enhance detection precision further we created a dual branch CNN model that leveraged layers to capture local and global features of the ECG signals. This dual branch model underwent. Validation using a dataset generated through SMOTE to tackle the inherent class imbalance in the dataset. Our model exhibited performance levels with accuracy rates and strong ROC AUC scores on both validation and test datasets. Additionally, we developed representations, like confusion matrices ROC curves and classification reports to assess the model’s effectiveness. In general, our research highlights how ECG driven detection systems, improved by DL and ML techniques, can offer a adaptable approach for identifying OSA. This solution could be incorporated into gadgets or household monitoring setups, for evaluation and timely identification.

  1. The study does not address explainability and interpretability, which are critical for AI models in healthcare applications.

Author’s reply

To further validate the model’s interpretive, Grad-CAM visualization was employed. Grad-CAM clearly identified the regions of the ECG that are most influential in the model’s prediction, highlighting key morphological features such as the QRS complexes and specific intervals that are critical for apnea detection. The brightly colored areas of the Grad-CAM visualization indicate segments the convolutional layers emphasized, providing crucial insight into the model’s decision-making process. The model's capacity to focus accurately on clinically relevant waveform features greatly enhances its interpretability and clinical trustworthiness, such that clinicians can have confidence in the underlying rationale for the predictions made (refer Fig 9).

Figure 9. Grad-CAM and Gradient

For improving clinical trust and interpretability, the model explainability methods such as Integrated Gradients and Grad-CAM were used. The Integrated Gradients analysis found that ECG waveform components like QRS complexes and the waveform intervals were the most important factors in predicting apnea events. Grad-CAM gave a visual proof of it by showing the areas of the ECG signals which were important for the convolutional layers and hence were important features and these features were in sync with the clinical understanding (Refer Fig 6). These results thus have a high clinical acceptance relevance as they show that the model's decisions are in close relation to physiological patterns and clinical reasoning. Therefore, these results clearly support the validity of the model for reliable, clinically relevant apnea detection and could be positioned as a promising tool for assisting healthcare professionals in clinical practice.

Table 3. Classification report of CNN model under train, test and evaluation tests.

Dataset

Class

Precision

Recall

F1-Score

Train

0

0.98

0.96

0.97

1

0.96

0.98

0.97

Validation

0

0.98

0.89

0.93

1

0.90

0.98

0.94

Test

0

0.94

0.94

0.94

1

0.94

0.93

0.94

Figure 4. ROC AUC graph of train, test and validation of CNN model.

Figure 5. Confusion Matrix of CNN model for test set.

Figure 6. Grad-CAM and Gradient

  1. The manuscript contains grammatical errors and lacks clarity in multiple sections, affecting readability and comprehension.

Author’s reply

We appreciate the reviewer’s feedback and acknowledge the importance of clarity and grammatical accuracy in scientific writing.

  1. The discussion and conclusion sections are overly optimistic and fail to critically analyze potential drawbacks.

Author’s reply

Nevertheless, the challenges of overfitting detected in models like Decision Trees and the struggles in identifying non apnea cases particularly with RF models emphasize the necessity for ongoing refinement and validation. Addressing these obstacles will necessitate exploration and advancement including incorporating more varied datasets, improving handling of class imbalances and possibly integrating hybrid models that merge different algorithm strengths.

  1. The manuscript does not meet the standard criteria for publication due to some issues in experimental rigor and result validation

Author’s reply

  1. check reference 21.

Updated

  1. Discuss the parameters for Gaussian noise injected during data augmentation.

Author’s reply

Data Augmentation

To make the CNN model more robust, in predicting apnea events from ECG signals data augmentation methods were employed. These techniques involved exposing the model to a range of signal patterns, such as simulating heart rates through time stretching and mimicking sensor sensitivities by adjusting amplitude scaling. Sensor noise was replicated using Gaussian noise and variability in recording lengths was introduced through cropping. By incorporating these strategies, the training data became more varied resulting in improved performance when processing ECG signals, in real world scenarios.

  1. Table 1. Summary of the Convolutional Neural Network (CNN) Model Architecture.. present it in professional manner. data from the python environment is pasted as table. what is 'none' here.

Author’s reply

  1. same for 'Table 2. Summary of the Dual Branch Model Architecture'.

Author’s reply

Updated

  1. check table 3. first part is for training?

Author’s reply

Updated the concern, we would like to thank all the reviewers.

  1. Figure 5. Confusion Matrix of CNN model for test set. Data set is very imbalanced. 

Author’s reply

Updated the figures

  1. check the references and citations.\

Author’s reply

Updated the citations and references

Reviewer 2 Report

Comments and Suggestions for Authors

The paper presents a method for detecting Obstructive Sleep Apnea (OSA) using ECG data. While the results appear promising, several issues need to be addressed before publication:

1. The paper contains numerous typos and poorly structured sentences (e.g., lines 22–23, line 290). The writing suggests it may have been translated from another language without sufficient revision. Professional English proofreading is recommended.

2. Several abbreviations are not defined upon first use, and some datasets mentioned lack proper references. Ensure that every abbreviation is introduced when first used and that each dataset is appropriately cited.

2. The current literature review consists of a collection of summarized papers without a clear connection to the research topic. Notably, reference 3 does not seem relevant to automated diagnosis. Consider rewriting this section, integrating it into the introduction to highlight the limitations of existing methodologies and justify the proposed approach.

3. Convolutional Neural Networks (CNNs) exhibit stochastic behavior. To ensure reproducibility, run the algorithm multiple times (e.g., 50+ iterations) within an X-fold cross-validation framework. Report performance metrics as mean ± standard deviation to demonstrate consistency.

4. Achieving near-perfect validation performance across all models raises concerns about a potential methodological flaw, particularly regarding data splitting. Ensure that data augmentation is performed after splitting the dataset to avoid information leakage.

5. The results and discussion sections should be better separated. Avoid repeating tables within the text; instead, focus on summarizing and generalizing the findings to improve readability.

6. The conclusions should be more concise, emphasizing the key contributions and findings of the study. Avoid unnecessary repetition and ensure the section clearly highlights the study's achievements.

Addressing these concerns will significantly improve the clarity, rigor, and scientific validity of the manuscript.

Comments on the Quality of English Language

Extensive English proofreading is necessarily.

Author Response

Rev-2

Comments and Suggestions for Authors

The paper presents a method for detecting Obstructive Sleep Apnea (OSA) using ECG data. While the results appear promising, several issues need to be addressed before publication:

Author’s reply

We appreciate the reviewer’s valuable feedback and have carefully revised the manuscript to emphasize the novelty of our study. To address this concern, we have made the following modifications:

  1. The paper contains numerous typos and poorly structured sentences (e.g., lines 22–23, line 290). The writing suggests it may have been translated from another language without sufficient revision. Professional English proofreading is recommended.

Author’s reply

We appreciate the reviewer’s valuable feedback and have carefully revised the manuscript to emphasize the novelty of our study.

Several abbreviations are not defined upon first use, and some datasets mentioned lack proper references. Ensure that every abbreviation is introduced when first used and that each dataset is appropriately cited.

Author’s reply

We appreciate the reviewer’s valuable feedback and have carefully revised the manuscript to emphasize the novelty of our study.

The current literature review consists of a collection of summarized papers without a clear connection to the research topic. Notably, reference 3 does not seem relevant to automated diagnosis. Consider rewriting this section, integrating it into the introduction to highlight the limitations of existing methodologies and justify the proposed approach.

Author’s reply

We appreciate the reviewer’s valuable feedback and have carefully revised the manuscript to emphasize the novelty of our study.

  1. Convolutional Neural Networks (CNNs) exhibit stochastic behavior. To ensure reproducibility, run the algorithm multiple times (e.g., 50+ iterations) within an X-fold cross-validation framework. Report performance metrics as mean ± standard deviation to demonstrate consistency.

Author’s reply

We appreciate the reviewer’s valuable feedback and have carefully revised the manuscript to emphasize the novelty of our study.

We have added these results in the MS

Performance Metrics (Mean ± Std) across 5 folds:

Dataset

Class

Precision

Recall

F1-Score

Accuracy

Train

0

0.97 ± 0.03

0.97 ± 0.01

0.97 ± 0.02

0.97 ± 0.02

1

0.97 ± 0.01

0.96 ± 0.04

0.97 ± 0.02

Validation

0

0.97 ± 0.05

0.95 ± 0.03

0.96 ± 0.03

0.96 ± 0.03

1

0.96 ± 0.02

0.97 ± 0.05

0.96 ± 0.03

Test

0

0.95 ± 0.05

0.95 ± 0.03

0.95 ± 0.03

0.95 ± 0.03

1

0.95 ± 0.02

0.95 ± 0.05

0.95 ± 0.03

  1. Achieving near-perfect validation performance across all models raises concerns about a potential methodological flaw, particularly regarding data splitting. Ensure that data augmentation is performed after splitting the dataset to avoid information leakage.

Author’s reply

Data Augmentation

To make the CNN model more robust, in predicting apnea events from ECG signals data augmentation methods were employed. These techniques involved exposing the model to a range of signal patterns, such as simulating heart rates through time stretching and mimicking sensor sensitivities by adjusting amplitude scaling. Sensor noise was replicated using Gaussian noise and variability in recording lengths was introduced through cropping. By incorporating these strategies, the training data became more varied resulting in improved performance when processing ECG signals, in real world scenarios.

  1. The results and discussion sections should be separated better. Avoid repeating tables within the text; instead, focus on summarizing and generalizing the findings to improve readability.

Author’s reply

Now the updated ms is readable

  1. The conclusions should be more concise, emphasizing the key contributions and findings of the study. Avoid unnecessary repetition and ensure the section clearly highlights the study's achievements.

Addressing these concerns will significantly improve the clarity, rigor, and scientific validity of the manuscript.

Author’s reply

We appreciate the reviewer’s valuable feedback and have carefully revised the manuscript to emphasize the novelty of our study.

Reviewer 3 Report

Comments and Suggestions for Authors

Reviewer’s report on “AI-driven detection of obstructive sleep Apnea using dual branch CNN and machine learning models”

The paper proposes a deep learning method based on a dual-branch CNN to detect obstructive sleep apnea (OSA) using electrocardiogram (ECG) data. The study is interesting and has a significant impact on public health and well-being. However, several key issues must be addressed before the paper can progress further.

First, the abstract should be written in the present tense and can be slightly shortened. Additionally, the English writing must be significantly improved. There are several typos, such as the phrase “These models were. Validated using,” which needs correction.

Second, the contribution of the research should be better explained. How does this study differ from existing published research, and what is novel about it? These aspects are not clearly articulated in the current manuscript.  Furthermore, the literature review section appears somewhat disorganized. It would be beneficial to clearly outline the main findings of each paper, along with their limitations and the ML/DL techniques used.

The methodology of the research (Figure 1) requires further explanation. How was this methodology developed? What feature extraction method was used? How was noise in the data removed? Additionally, a detailed breakdown of the steps involved should be provided.  What is the source of the data? Was it obtained from a publicly available dataset or a private source?  How much did data augmentation improve the resolution of the data and the accuracy of the results? Additionally, the subsections in Section 3 must be numbered, as the current structure makes it difficult to follow the flow of content.

Section 4 is the most interesting part of the paper, as it compares the results obtained from different ML/DL techniques. Summarizing all the comparison results in a single table would be beneficial, allowing readers to better understand the findings.

Comments on the Quality of English Language

As above

Author Response

Rev-3

Comments and Suggestions for Authors

Reviewer’s report on “AI-driven detection of obstructive sleep Apnea using dual branch CNN and machine learning models”

The paper proposes a deep learning method based on a dual-branch CNN to detect obstructive sleep apnea (OSA) using electrocardiogram (ECG) data. The study is interesting and has a significant impact on public health and well-being. However, several key issues must be addressed before the paper can progress further.

Author’s reply

We appreciate the reviewer’s valuable feedback and have carefully revised the manuscript to emphasize the novelty of our study.

First, the abstract should be written in the present tense and can be slightly shortened. Additionally, the English writing must be significantly improved. There are several typos, such as the phrase “These models were. Validated using,” which needs correction.

Author’s reply

We have updated the abstract accordingly

Second, the contribution of the research should be better explained. How does this study differ from existing published research, and what is novel about it? These aspects are not clearly articulated in the current manuscript.  Furthermore, the literature review section appears somewhat disorganized. It would be beneficial to clearly outline the main findings of each paper, along with their limitations and the ML/DL techniques used.

Author’s reply

We have updated the MS with respect to contribution and literature accordingly

The methodology of the research (Figure 1) requires further explanation. How was this methodology developed? What feature extraction method was used? How was noise in the data removed? Additionally, a detailed breakdown of the steps involved should be provided.  What is the source of the data? Was it obtained from a publicly available dataset or a private source?  How much did data augmentation improves the resolution of the data and the accuracy of the results? Additionally, the subsections in Section 3 must be numbered, as the current structure makes it difficult to follow the flow of content.

Author’s reply

We have updated the MS as mentioned in the reviewer’s report

Section 4 is the most interesting part of the paper, as it compares the results obtained from different ML/DL techniques. Summarizing all the comparison results in a single table would be beneficial, allowing readers to better understand the findings.

We have updated the MS as mentioned in the reviewer’s report

Round 2

Reviewer 1 Report

Comments and Suggestions for Authors

Most of the commnets are adressed. However, comaprision with other state of the art algorithms in the area is required.

Comments on the Quality of English Language

Most of the commnets are adressed. However, comaprision with other state of the art algorithms in the area is required.

Author Response

Reviewer-1

Most of the comments are addressed. However, comparison with other state of the art algorithms in the area is required.

Author’s reply

We completely agree with the reviewer’s point of view; therefore, we have added the following information. Secondly, opted native speaker to proofread our article.

As shown in Fig 15, authors presents [13] an approach, to identifying sleep apnea (SLA) by analyzing single lead ECG signals. It uses a Deep Belief Network (E DBN) incorporating two types of Restricted Boltzmann Machines (RBM). After being trained on the Apnea ECG dataset the model achieved an accuracy of 89.11% per segment and 97.17% per recording offering a affordable option compared to polysomnography tests. In [14], identifying sleep apnea by analyzing ECG signals using a neural network (DNN) that incorporates LSTM and BiLSTM models. The BiLSTM model, which utilizes backward learning attained an accuracy rate on the Physionet dataset showcasing exceptional capability, in detecting sleep apnea.

Figure 15. Comparison with state-of-the-art literature

Reviewer 2 Report

Comments and Suggestions for Authors

Second round of review
Previous 1. The paper has improved but Professional English proofreading is recommended.
Previous 3. Convolutional Neural Networks (CNNs) exhibit stochastic behavior. To ensure reproducibility, run the algorithm multiple times (e.g., 50+ iterations) within an X-fold cross-validation framework. Report performance metrics as mean ± standard deviation to demonstrate consistency.
This was not properly addressed. It should be applied to the both algorithms, explained in the methods, and reported in the results.
Previous 4. Achieving near-perfect validation performance across all models raises concerns about a potential methodological flaw, particularly regarding data splitting. Ensure that data augmentation is performed after splitting the dataset to avoid information leakage.
This has not been properly addressed. Clear demarcation of split before augmentation should be implemented and clearly stated.
6. The conclusions should be more concise, emphasizing the key contributions and findings of the study. Avoid unnecessary repetition and ensure the section clearly highlights the study's achievements.
This has not been properly addressed. Please highlight your contributions and not general aspects. The paragraph added seems more like a limitation subchapter for discussion.
Addressing these concerns will significantly improve the clarity, rigor, and scientific validity of the manuscript.
More precise response to concerns should be implemented.

Comments on the Quality of English Language

The paper needs professional proofreading (also academic).

Author Response

Second round of review
Previous 1. The paper has improved but Professional English proofreading is recommended.

Authors Reply

We have opted English service for this paper.

Second round of review
Previous 3. Convolutional Neural Networks (CNNs) exhibit stochastic behavior. To ensure reproducibility, run the algorithm multiple times (e.g., 50+ iterations) within an X-fold cross-validation framework. Report performance metrics as mean ± standard deviation to demonstrate consistency.
This was not properly addressed. It should be applied to both algorithms, explained in the methods, and reported in the results.

Authors Reply

We ran our experiment as mentioned by the respected reviewer and we have received the following results for these two models. These results are mentioned in the article as well.

Model Name

Mean ± standard deviation

CNN

Final Cross-Validation Results:

Average ROC-AUC: 0.9487 ± 0.0274

Average Accuracy: 0.8783 ± 0.0378

Dual Branch CNN

Average ROC-AUC: 0.9019 ± 0.0271 Average Accuracy: 0.896 ± 0.138

Second round of review
Previous 4. Achieving near-perfect validation performance across all models raises concerns about a potential methodological flaw, particularly regarding data splitting. Ensure that data augmentation is performed after splitting the dataset to avoid information leakage.
This has not been properly addressed. Clear demarcation of split before augmentation should be implemented and clearly stated.

Authors Reply

We have added this paragraph to the data augmentation section

Our approach was intended to avoid information leakage. Therefore, we have divided the dataset into training, testing and validation before any preprocessing and augmentation. Data augmentation applied to the training dataset only to address class imbalance, while both test and validation sets were never touched or augmented. These settings enabled us to calculate the classification threshold using prediction from the training set and then only applied to test and validation sets. Therefore, the model guarantees unbiased and reliable performance metrics by keeping the datasets strictly separate and using them only for their intended purposes, to prevent any leakage of information that could artificially inflate results.

Second round of review
6. The conclusions should be more concise, emphasizing the key contributions and findings of the study. Avoid unnecessary repetition and ensure the section clearly highlights the study's achievements.
This has not been properly addressed. Please highlight your contributions and not general aspects. The paragraph added seems more like a limitation subchapter for discussion.
Addressing these concerns will significantly improve the clarity, rigor, and scientific validity of the manuscript.
More precise responses to concerns should be implemented.

Authors Reply

We have updated the conclusion section, please find the below text for the same.

The current study establishes the possibility and efficacy of using sophisticated DL techniques such as CNNs and Dual-Branch CNNs for identifying sleep apnea from ECG signals. These techniques are superior to traditional methods, such as DT and RF, in terms of accuracy and precision. The Dual-Branch CNN was found to have a greater ability to detect the complicated ECG signals associated with apnea events, thus validating the effectiveness of such specialized neural network architectures for health informatics. However, both DT and RF models have drawbacks. They are likely to overfit the data and are not very accurate when applied to non-apnea instances. Future work should address the model limitations, for example, by using diverse datasets, improving the handling of class imbalance, and investigating hybrid modeling strategies to improve diagnostic accuracy.

Reviewer 3 Report

Comments and Suggestions for Authors

The authors should provide a detailed response letter addressing each comment point by point. Unfortunately, the current version does not include specific answers; instead, it merely directs the reviewers to check the updated manuscript. A proper response letter should clearly outline the changes made, explain how each concern has been addressed, and provide justification for any suggestions that were not implemented.

Author Response

Second round of review

Reviewer’s report on “AI-driven detection of obstructive sleep Apnea using dual branch CNN and machine learning models”

Second round of review

The paper proposes a deep learning method based on a dual-branch CNN to detect obstructive sleep apnea (OSA) using electrocardiogram (ECG) data. The study is interesting and has a significant impact on public health and well-being. However, several key issues must be addressed before the paper can progress further.

Authors Reply

We sincerely thank the reviewers for their insightful comments and constructive feedback on our manuscript. We appreciate the recognition of the study's potential impact on public health and well-being. Below, we address the key issues raised by the reviewers and outline the steps we have taken to improve the manuscript.

Second round of review

First, the abstract should be written in the present tense and can be slightly shortened. Additionally, the English writing must be significantly improved. There are several typos, such as the phrase “These models were. Validated using,” which needs correction.

Authors Reply

The updated abstract is as follows. We would like to thank our reviewers for tier efforts in making this MS more readable.

The purpose of this research is to compare and contrast the application of machine learning and deep learning methodologies such as a Dual Branch Convolutional Neural Network (CNN) model for detecting Obstructive Sleep Apnea (OSA) from electrocardiogram (ECG) data. Therefore, this approach solves the limitations of conventional polysomnography (PSG) and presents a non-invasive method of detecting OSA in its early stages with the help of AI. The research shows that both CNN and Dual Branch CNN models can identify OSA from ECG signals. The CNN model achieves validation and test accuracy of about 93% and 94%, respectively. Whereas the dual branch CNN model has achieved 93% validation and 94% test accuracy.   Furthermore, the Dual Branch CNN obtains a ROC AUC score of 0.99, which is better at distinguishing between apnea and non-apnea cases. The results show that CNN models, especially the Dual Branch CNN, are effective in apnea classification and better than traditional methods. In addition, our proposed model has the potential to be a reliable non-invasive method for accurate OSA detection that is even better than the current state-of-the-art advanced methods.

Second round of review

Second, the contribution of the research should be better explained. How does this study differ from existing published research, and what is novel about it? These aspects are not clearly articulated in the current manuscript.  Furthermore, the literature review section appears somewhat disorganized. It would be beneficial to clearly outline the main findings of each paper, along with their limitations and the ML/DL techniques used.

Authors Reply

We sincerely thank the reviewer for their valuable feedback. We acknowledge that the contribution of the research and organization of the literature review section could be improved. Below, we address these concerns in detail. The following text is added at the literature review section. The following table is added the literature review section.

  Table 1. Summary of state-of-the-art techniques, Accomplishments, and issues in Obstructive Sleep Apnea (OSA) Detection

Reference

Technique Used

Achievements

Problems Identified

[1] Sheta et al. (2021)

Dual Branch CNN on ECG signals

Higher accuracy

No significant drawbacks mentioned

[2] Bernardini et al. (2021)

AIOSA Deep Learning Model (Heart & breathing signals)

Higher accuracy

Computationally expensive; sensitive to signal noise; lack of explainability

[3] Kim et al. (2021)

KL-6 biomarker blood test

identified OSA severity

Invasive method

[4] Huang et al. (2024)

Logistic Regression with BMI and TyG index

Low AUC

Limited validation

[5] Li et al. (2024)

ResNeSt34 Deep Learning (ECG + Thoracic signals)

High Accuracy

Works well with the time series text data but reduced performance when used with ECG

[6] Yue et al. (2021)

Multi-resolution Residual Network (Mr-ResNet) with nasal airflow

Excellent accuracy, sensitivity,

Invasive method

[7] Monna et al. (2022)

3D Maxillofacial Shape Analysis and Machine Learning

High Accuracy

3d Scanning is needed, it means extra load on the CPU 

[8] Cen et al. (2018)

CNN

Real-time OSA event detection accuracy ~79.61%

Complex, has low performance when signals are mixed

[9] Nassehi et al. (2024)

KNN classifier on resting state EEG

High accuracy (93.33%), excellent AUC (0.98)

Dataset is very small

[10] Liu et al. (2024)

EfficientNet + XGBoost on single-lead ECG

Excellent AUC (0.917-0.975), high accuracy (85.5%-92.8%)

Model interpretation focuses only on specific apnea events, hence model has difficulty in identifying hypopneas with arousals

[11] Jiménez-García et al. (2024)

CNN + RNN on airflow and oximetry signals (pediatric)

High diagnostic accuracy (≥84%), good interpretability using Grad-CAM

Only for pediatrics

Many current studies employ small manual or semi-automated annotations and use standard signal preprocessing and augmentation techniques to complement the data. For instance, good accuracy has been achieved by standard CNN models, but they have shown restricted generalizability to unannotated regions with dataset specific annotations and poor signal quality.  Furthermore, previous research on machine generated QRS annotations often involved inaccuracies that needed manual correction, which may have compromised the reliability of the models.

On the other hand, our study makes several key contributions to improve apnea detection from ECG signals. We use dual-branch CNN architecture to train both broad and fine-grained ECG  signal features simultaneously and are thus more robust than CNN models. Furthermore, our rigorous preprocessing pipeline entails sophisticated bandpass filtering, stringent segment normalization, and outlier detection to clean and stabilize the input data.

Furthermore, we have proposed a new algorithm for annotation of ECGs by simply counting the number of QRS complexes in an ECG segment, which increases the annotation accuracy and decreases the dependence on the inaccurate machine generated labels. Our method is unique in that it clearly separates training, validation, and testing processes to produce clinically meaningful and accurate results. We hereby propose collective enhancements as the novelty of  our study and distinguish it from other literature.

Reference [1], they developed a new annotation method aimed at improving the accuracy of ECG-based detection of OSA. They introduced a robust Dual Branch CNN model tailored for this purpose, which performed better than existing advanced approaches. Their technique provides a reliable, precise, and non-invasive diagnostic solution with considerable potential for practical application in clinical environments [1]. AIOSA is a deep learning model developed to detect OSA from heart and breathing signals. It achieves high accuracy and is compatible with data collected from hospitals and standard benchmarks. However, it is computationally expensive, sensitive to signal noise, and its functioning is not easily explainable which poses a challenge in accepting the model in clinical practice to some extent [2]. The researchers tried to see if KL-6 levels in the blood could reveal ‘occult’ lung injury in people with OSA. The authors studied 197 patients and observed that higher KL-6 levels were clearly related to more severe OSA [3]. The research [4] aimed to develop a user screening model, for OSA using biochemistry markers and demographic information. Data from the years 2020 to 2023 were examined using four machine learning techniques with Logistic Regression (LR) showing the performance. The LR model, including factors such as BMI (AUC = 0.699) and the glucose (TyG) index (AUC = 0.656) demonstrated results in training (AUC = 0.794 F1 score = 0.841) validation (AUC = 0.777 F1 score = 0.827) and testing groups (AUC = 0.732 F1 score = 0.788). This model serves as a tool for healthcare professionals to assess the need for PSG, in suspected OSA patients. Top of Form The model performance is quite good but has not been validated in any other patient group other than the one used in the original study, hence one cannot have much confidence in its usefulness in other populations or healthcare contexts.

The research [5] aimed to create an automated deep learning system to identify OSA events by analyzing both ECG and thoracic movement signals together. By using data from 420 PSG cases the model was built using the ResNeSt34 algorithm. The combined signals approach outperformed the ECG method achieving an accuracy rate of 89.0% precision of 88.8% recall of 89.0%, F1 score of 88.2% and AUC of 92.9%. In comparison the ECG model had an accurate rate of 84.1% precision of 83.1% recall of 84.1%, F1 score of 83.3% and AUC of 82.8%. These findings highlight the effectiveness when incorporating thoracic movement signals, in detecting OSA events. Authors [6] evaluated the Obstructive Sleep Apnea Smart System (OSASS), which uses a novel multi-resolution residual network (Mr-ResNet) to diagnose and classify OSA based on single-channel nasal pressure airflow signals. Data from sleep centers were used to develop and test the model. In the primary test set, Mr-ResNet achieved sensitivity, specificity, accuracy, and F1-score of 90.8%, 90.5%, 91.2%, and 90.5%, respectively. OSASS showed strong agreement with registered polysomnographic technologists' (RPSGT) scores, with Spearman correlations of 0.94 and 0.96, and Cohen’s Kappa scores of 0.81 and 0.84. The findings suggest that OSASS could be a valuable tool for clinical OSA diagnosis and classification. The authors [7] explore the use of 3D maxillofacial shape analysis combined with machine learning (ML) to predict OSA in 280 Caucasian men. Compared to traditional questionnaires (BERLIN and NoSAS), the ML analysis of 3D craniofacial shapes demonstrated higher specificity (56%) and good sensitivity (80%) for detecting OSA (AHI≥15 events/h). The performance further improved (ROC = 0.75) when combined with patient anthropometrics. This approach is proposed as a rapid, efficient, and cost-effective OSA screening tool, offering a promising alternative to traditional methods PSG. The authors investigated the automatic detection of OSA events using a deep learning approach, specifically CNN. The model analyzes blood oxygen saturation, oronasal airflow, and ribcage and abdomen movements to classify sleep apnea events with 1-second annotations. Utilizing the PhysioNet Sleep Database, the model achieved an average accuracy of 79.61% across normal, hypopnea, and apnea classes, demonstrating the potential of CNNs in accurately detecting OSA events in real-time [8]. The researchers [9] delve into determining the severity of OSA by analyzing resting state EEG data without relying on PSG signals. The study examined EEG recordings from 25 OSA patients focusing on features derived from delta, theta and alpha sub-bands. Using ML techniques, the research identified combinations of features with the Relief method pinpointing 15 features. The K Nearest Neighbors classifier (, with K=5) demonstrated excellent performance metrics achieving an accuracy rate of 93.33%, sensitivity of 92.30% specificity of 94.14% and an AUC value of 0.98. These results indicate that utilizing resting state EEG could serve as an efficient method for evaluating OSA severity. The researchers [10] aimed to create a computer program using intelligence to recognize patterns of sleep apnea in ECG signals, from a lead. By analyzing data from 1,656 patients, the model Efficient Net was used to detect apnea and by adjusting slicing and sample weight settings the best outcomes were achieved (AUC; 0.917 accuracy; 0.855). When paired with XGBoost for screening patients with AHI > 30 the system reached an AUC of 0.975 and accuracy of 0.928. Experiments conducted on PhysioNet data validated the model’s effectiveness, in identifying levels of sleep apnea. The AHI regression showed agreement (intraclass correlation coefficient > 0.9) with OSA severity classification achieving accuracies of 74.51% and 62.31%, for the four class categories along with Cohen’s Kappa values of 0.6231 and 0.4495 in CHAT and private datasets. The diagnostic accuracies for AHI cutoffs (1, 5 and 10 events/h) were consistently above 84%. Analysis using Grad CAM heatmaps indicated that the model focuses on AF cessations and SpO2 drops to identify apneas and hypopneas with desaturations while often overlooking patterns of hypopneas associated with arousals. Therefore, employing a CNN + RNN model to assess AF and SpO2 could serve as a diagnostic tool for symptomatic children, at risk of OSA [11].

Second round of review

The methodology of the research (Figure 1) requires further explanation. How was this methodology developed? What feature extraction method was used? How was noise in the data removed? Additionally, a detailed breakdown of the steps involved should be provided.  What is the source of the data? Was it obtained from a publicly available dataset or a private source?  How much did data augmentation improve the resolution of the data and the accuracy of the results? Additionally, the subsections in Section 3 must be numbered, as the current structure makes it difficult to follow the flow of content.

Authors Reply

Dataset

The dataset consists of 70 ECG recordings, split into a training set of 35 records and a test set of 35 records. Each recording spans, from 7 to 10 hours in duration. It contains ECG signals, apnea annotations (only for the training set) and QRS annotations. In addition, eight records include signals (chest and abdominal effort oronasal airflow) and oxygen saturation (SpO2) data. This extensive dataset is designed for the development and testing of methods for apnea detection with files provided to aid in the examination of respiratory and ECG signals. It's worth noting that certain QRS files may contain errors that need correction, for analysis. Human generated labels created by experts to show whether apnea is present or not, on a minute-by-minute basis (only accessible, for the training dataset) [12]. The dataset offers two kinds of annotations for the computers, in Cardiology Challenge 2000 distinguished by the file extensions. apn. and qrs. The apn files include apnea annotations for the training set comprising 35 records established through expert assessment of respiration and oxygen saturation signals. Each annotation specifies whether apnea was ongoing at the beginning of each minute. The.qrs files, generated with the sqrs125 tool, they are machine generated annotations for QRS complexes. The rdann tool can also be employed to convert these annotation files into text for further examination.

Preprocessing

The initial steps outlined in this research are carefully crafted to prepare, standardize and harmonize the ECG data, for training a AI model. The process kicks off with bandpass filtering, which effectively eliminates any disturbances from the ECG signals (refer fig 2). Next the ECG signal is divided into segments to facilitate processing and analysis by the model. Normalization is then implemented to ensure uniformity of data across all segments, a factor for model performance (figure 3) Points, in the ECG signal that stand out are considered outliers showing deviations, from the amplitude. These outliers could suggest noise interference or unusual cardiac activities. It is essential to recognize and address these anomalies to guarantee ECG examination and understanding.

Figure 2. Comparison of original and smoothed ECG signals of the Apnea dataset.

Figure 3. Smoothed ECG Signal with Detected Outliers.

Feature extraction

We applied handcrafted feature extraction methods which derived meaningful information from the preprocessed signals. Our approach provided various steps, including loading of ECG data from two different files namely .hea and .dat. We have applied divides each signals and applied bandpass to remove noise. Secondly, we have derived temporal based information such as heart variability, statistical measures (mean, variance, skewness, kurtosis), and amplitude characteristics (max, min, peak-to-peak), can be computed from the filtered signal.  Furthermore, for the frequency domain features namely power spectral density and dominant frequency are derived using Fourier Transform. Time-frequency and its feature are able to capture both temporal and spectral changes.

CNNs are trained to extract relevant features from raw data. These models are designed to function without the need for explicit feature engineering, CNN models identify patterns and structures in the raw data. However, they are still useful, especially in conjunction with handcrafted features (e.g., time-domain, frequency-domain, or time-frequency features), which can both improve the performance of the OSA detection system above that of deep learning models on their own, by supplying handcrafted features to complement the learned features. This hybrid approach capitalizes on the strengths of both methods: deep learning for automated feature learning, and handcrafted features for capturing domain-specific insights.

Data Augmentation

Our approach was intended to avoid information leakage. Therefore, we have divided the dataset into training, testing and validation before any preprocessing and augmentation.  Data augmentation applied to the training dataset only to address class imbalance, while both test and validation sets were never touched or augmented. These settings enabled us to calculate the classification threshold using prediction from the training set and then only applied to test and validation sets. Therefore, the model guarantees unbiased and reliable performance metrics by keeping the datasets strictly separate and using them only for their intended purposes, to prevent any leakage of information that could artificially inflate results.

Second round of review

Section 4 is the most interesting part of the paper, as it compares the results obtained from different ML/DL techniques. Summarizing all the comparison results in a single table would be beneficial, allowing readers to better understand the findings.

Authors Reply

We are very thankful to the respected reviewers, because of their insightful comments made the MS more readable. Below text is added the text to give more understanding on the state of the art research.

As shown in Fig 15, authors presents [13] an approach, to identifying sleep apnea (SLA) by analyzing single lead ECG signals. It uses a Deep Belief Network (E DBN) incorporating two types of Restricted Boltzmann Machines (RBM). After being trained on the Apnea ECG dataset the model achieved an accuracy of 89.11% per segment and 97.17% per recording offering a affordable option compared to polysomnography tests. In [14], identifying sleep apnea by analyzing ECG signals using a neural network (DNN) that incorporates LSTM and BiLSTM models. The BiLSTM model, which utilizes backward learning attained an accuracy rate on the Physionet dataset showcasing exceptional capability, in detecting sleep apnea.

Figure 15. Comparison with state-of-the-art literature

Round 3

Reviewer 2 Report

Comments and Suggestions for Authors

The paper has improved.

The results presented in the Response 1, Performance Metrics (Mean ± Std) across 5 folds, should be introduced in the paper.

Comments on the Quality of English Language

The paper have improved, after introducing the 5-fold results in the paper, publication can be considered.

Author Response

Reviewers’ Comment

The results presented in the Response 1, Performance Metrics (Mean ± Std) across 5 folds, should be introduced in the paper.

Authors

We thank the reviewers for their insightful feedback on this methodology.

We created representations including confusion matrices ROC curves and classification reports to evaluate the model's effectiveness and understand classification results and provide unambiguous visual and quantitative insights into the model's performance across individual classes and the mean and standard deviation obtained across the five-fold cross-validation procedure.

Reviewer 3 Report

Comments and Suggestions for Authors

Thank you to the authors for addressing the reviewers’ comments to a satisfactory level. I am pleased with the corrections and amendments made in the revised manuscript, which have improved the clarity and overall quality of the paper. Therefore, I am happy to recommend the paper for acceptance.

Author Response

Reviewers; comment

Thank you to the authors for addressing the reviewers’ comments to a satisfactory level. I am pleased with the corrections and amendments made in the revised manuscript, which have improved the clarity and overall quality of the paper. Therefore, I am happy to recommend the paper for acceptance.

Author’s Comments

We sincerely thank the reviewers for their constructive feedback and valuable insights, which have significantly strengthened the manuscript. We are pleased that our revisions addressed all concerns to the reviewers' satisfaction, and we appreciate their recommendation for acceptance. The paper has benefited greatly from this collaborative review process.
